# Metabolic reprograming mediated by tumor cell-intrinsic type I IFN signaling is required for CD47-SIRPα blockade efficacy

Hang Zhou[1,5], Wenjun Wang[2,5] ✉, Hairong Xu[1], Yong Liang[3], Jiyu Ding[1,3], Mengjie Lv[2], Boyang Ren[1], Hua Peng [1], Yang-Xin Fu [4] ✉ & Mingzhao Zhu [1,2,3] ✉

Type I interferons have been well recognized for their roles in various types of immune cells during tumor immunotherapy. However, their direct effects on tumor cells are less understood. Oxidative phosphorylation is typically latent in tumor cells. Whether oxidative phosphorylation can be targeted for immunotherapy remains unclear. Here, we find that tumor cell responsiveness to type I, but not type II interferons, is essential for CD47-SIRPα blockade immunotherapy in female mice. Mechanistically, type I interferons directly reprogram tumor cell metabolism by activating oxidative phosphorylation for ATP production in an ISG15-dependent manner. ATP extracellular release is also promoted by type I interferons due to enhanced secretory autophagy. Functionally, tumor cells with genetic deficiency in oxidative phosphorylation or autophagy are resistant to CD47-SIRPα blockade. ATP released upon CD47-SIRPα blockade is required for antitumor T cell response induction via P2X7 receptor-mediated dendritic cell activation. Based on this mechanism, combinations with inhibitors of ATP-degrading ectoenzymes, CD39 and CD73, are designed and show synergistic antitumor effects with CD47-SIRPα blockade. Together, these data reveal an important role of type I interferons on tumor cell metabolic reprograming for tumor immunotherapy and provide rational strategies harnessing this mechanism for enhanced efficacy of CD47-SIRPα blockade.

Type I interferons (IFN-Is) are central organizers in tumor immunity[1,2]. Via coordinating various types of innate and adaptive immune cells, IFN-Is play a fundamental role in antitumor immunity, which has been well-established through numerous studies, including conventional chemotherapy, targeted chemotherapy, radiotherapy and immunotherapy[1–4]. However, how tumor cells respond to IFN-Is and to what extent this would influence antitumor immunity and therapeutic efficacy are still poorly understood. Recently, a seminal work[5], using a murine fibrosarcoma model, showed that the therapeutic activity of doxorubicin was highly dependent on IFN-I responsiveness of tumor cells instead of host cells. Mechanistically, IFN-I signaling in tumor cells upregulates expression of CXCL10, an important chemokine for effector T cell recruitment[6]. However, whether tumor cell responsiveness to IFN-I signaling plays an important role during immunotherapy, such as immune checkpoint blockade (ICB), remains undefined.

[1]Key Laboratory of Epigenetic Regulation and Intervention, Institute of Biophysics, Chinese Academy of Sciences, Beijing, China. [2]CAS Key Laboratory of Pathogen Microbiology and Immunology, Institute of Microbiology, Chinese Academy of Sciences, Beijing, China. [3]College of Life Sciences, University of the Chinese Academy of Sciences, Beijing, China. [4]Department of Basic Medical Sciences, School of Medicine, Tsinghua University, Beijing, China. [5]These authors contributed equally: Hang Zhou, Wenjun Wang. ✉e-mail: wangwenjun@im.ac.cn; yangxinfu@tsinghua.edu.cn; zhumz@im.ac.cn

In recent years, innate immune checkpoints have emerged as a type of popular targets for tumor immunotherapy[7,8]. CD47-SIRPα axis is the first checkpoint to show that blockade of which inhibits growth of both blood cancers and solid tumors[9–12]. Therapies designed to disrupt CD47-SIRPα interaction have since been tested in numerous preclinical and clinical studies[13]. However, while CD47-SIRPα blockade showed impressive efficacy in some animal models, clinical efficacy is limited and especially poor for solid tumors[7,14,15]. Just recently, a late clinical trial of a CD47-blocking antibody was discontinued, and some other trials were also restrained[16]. Thus, at this time point, whether CD47-SIRPα blockade strategy can still be exploited for clinical advantage is facing a challenge. Further understanding the underlying mechanisms is urgently needed to enhance its efficacy or predict for accurate medicine. Our previous animal studies have shown that IFN-I signaling plays a critical role in CD47-SIRPα blockade therapy[17]. Upon CD47-SIRPα blockade, an ample amount of IFN-Is is produced from myeloid cells[17], and required for the induction of antitumor T cell response[17]. Lack of IFNAR1 in dendritic cells (DCs) significantly blunted the antitumor T cell response and efficacy of CD47-SIRPα blockade[17], suggesting an essential role of IFN-I signaling in DCs. However, it is unknown whether tumor cell responsiveness to IFN-Is is also required for the therapeutic efficacy.

A hallmark of tumor cells is altered metabolism, usually demonstrated as a predominant Warburg effect, and reduced mitochondrial oxidative phosphorylation (OXPHOS)[18]. This is not due to permanent impairment of mitochondrial OXPHOS, but partly due to suppression by enhanced glycolysis[19–24]. In fact, tumor cell metabolism is plastic and can be further reprogrammed in response to external stimuli[25,26]. In the past decade, the role of the Warburg effect in tumor immunity has been extensively studied. As a primary product of the Warburg effect, lactic acid accumulation and the resulting acidic tumor microenvironment (TME) have been well documented to inhibit both innate and adaptive antitumor immunity[27–34]. In contrast, the function of OXPHOS in antitumor immunity remains barely understood. Products of OXPHOS, such as ATP and ROS, have demonstrated important functions in promoting antitumor responses[35–38]. It is therefore possible that if tumor metabolism can be reprogrammed toward OXPHOS properly, it might enhance antitumor immunity. However, given the direct role of mitochondrial OXPHOS in tumor oncogenesis[39–41], metastasis[42,43] and chemo- or radiotherapy-resistance[44,45], inhibition of tumor cell mitochondrial OXPHOS was frequently proposed in tumor therapy[46–48]. Thus, in the context of immunotherapy, the role of tumor cell OXPHOS remains to be elucidated.

In this work, using IFN-I and IFN-II receptor knockout tumor models, we evaluate the role of IFN responsiveness of tumor cells during CD47-SIRPα blockade therapy. Interestingly, we find that the therapeutic efficacy of CD47-SIRPα blockade is highly dependent on tumor cell-intrinsic IFN-I signaling, but not IFN-II signaling. Mechanistically, IFN-Is, but not IFN-II, are found to both reprogram tumor cell metabolism toward OXPHOS for ATP production and induce tumor cell autophagy for ATP extracellular release. Further investigations show that ATP released upon CD47-SIRPα blockade is required for antitumor T cell response induction via P2X7 receptor-mediated DC activation. Combination with inhibitors of CD39 or CD73, the prevalent degrading ectoenzymes of ATP in TME, leads to synergistic effects with CD47-SIRPα blockade. Thus, our study reveals an important role of IFN-I in reprograming tumor cell metabolism for antitumor immunity during CD47-SIRPα blockade and highlights the importance of tumor cell OXPHOS in immunotherapy.

## Results
### Tumor cell-intrinsic IFN-I signaling is essential for CD47-SIRPα blockade therapy in mice
To investigate tumor cell responsiveness during immunotherapy, established murine MC38 tumors were treated with mutant SIRPα- Fc

(hIgG1) fusion protein (CV-1)[49,50], a high-affinity reagent for CD47 blockade or control hIgG. Tumor cells were subsequently sorted for RNA-seq analysis after three treatments (Fig. 1a). Differentially expressed genes were analyzed for Gene Ontology. The results showed that pathways associated with cellular response to IFN-I and IFN-II were significantly enriched (Fig. 1b). To determine whether tumor cell-intrinsic IFN signaling pathways might contribute to the therapeutic effect of CV-1, IFN-I receptor (IFNAR1)-knockout (KO) and IFN-II receptor (IFNGR1) KO MC38 cell lines were constructed using CRISPR-Cas9 (Supplementary Fig. 1a, b). The absence of STAT1 activation and ISGs upregulation in IFNAR1 and IFNGR1 KO cells upon IFN-α or IFN-γ treatment, respectively, further confirmed the knockout efficiency (Supplementary Fig. 1c, d). Upon CV-1 treatment, WT MC38 tumor growth was significantly inhibited and mouse survival rate was also increased compared to control hIgG treatment (Fig. 1c, f). In contrast, IFNAR1 KO MC38 tumors were resistant to CV-1 treatment and tumor-bearing mice showed no benefit of survival (Fig. 1d, g). IFNGR1 KO MC38 tumors remain responsive to CV-1 treatment (Fig. 1e, h). Similar results were observed in mice treated with Miap301[17], a CD47 blocking antibody (Supplementary Fig. 2a–c). In addition to MC38, CT26 (Supplementary Fig. 1c, d, and Supplementary Fig. 2d–i) and A20 (Supplementary Fig. 1c, d, and Supplementary Fig. 2j–o) tumor models also demonstrated the essential role of tumor cell-intrinsic IFN-I signaling, but not IFN-II signaling, in the context of CD47-SIRPα blockade therapy. Thus, these data indicate that IFN-I pathway is activated in tumor cells during CD47-SIRPα blockade treatment, and that tumor cell-intrinsic type I but not type II IFN signaling is essential for the therapeutic effect of CD47-SIRPα blockade therapy in mice.

### IFN-Is promote ATP release from tumor cells which is essential for CD47-SIRPα blockade therapy in mice
Next, we wondered how the activated IFN-I signaling pathway affects tumor cells in response to CD47-SIRPα blockade. IFN-α hardly inhibited MC38 tumor cell proliferation and minimally induced apoptosis in vitro (Supplementary Fig. 3a, b). The slight effect of IFN-α on MC38 proliferation/apoptosis does not appear to be the major factor conferring tumor growth inhibition in vivo, as tumor growth inhibition was totally abolished in T cell deficient mice or upon CD8 + T cell depletion (Fig. 2a, b), suggesting the involvement of T cell response. Consistent with this, CV-1 treatment induced significantly more tumor infiltrating tumor antigen-specific CD8 + T cells than control hIgG treatment (Fig. 2c, d). This suggests that IFN-I signaling in tumor cells may shape antitumor T cell immunity. We then asked whether IFN-Is might induce immunogenic apoptosis[51,52], during which released damage-associated molecular patterns (DAMPs) could facilitate antitumor T cell response. Among the major DAMP molecules, ATP was markedly released from MC38 cells upon IFN-α treatment in vitro (Fig. 2e), while the exposure of calreticulin to the plasma membrane was only slightly increased (Supplementary Fig. 3c). Marked release of ATP upon IFN-α treatment was also observed in A20 tumor cells, a murine lymphoma cell line, and HT29, a human colon adenocarcinoma cell line (Supplementary Fig. 3d, e). To further confirm the phenotype of ATP release in tumor tissues in vivo upon CD47-SIRPα blockade, we measured tumor tissue extracellular ATP using an ATP probe and the IVIS system. A significantly increased level of extracellular ATP concentration was observed in tumor tissues upon CD47-SIRPα blockade (Fig. 2f).

Extracellular ATP has been reported to play an important role in promoting antitumor immunity[35,36]. We further investigated whether ATP is involved in CD47-SIRPα blockade therapy. Co-administration of PPADS, an ATP receptor antagonist[53], indeed significantly inhibited the therapeutic effect of CV-1 in WT MC38 tumors (Fig. 2g), indicating the essential role of ATP in CV-1 treatment. For IFNAR1 KO MC38 tumors, while single CV-1 treatment had no therapeutic effect, additional injection of BzATP, a P2X7 receptor agonist[54], significantly inhibited tumor growth, further suggesting that ATP is an important factor

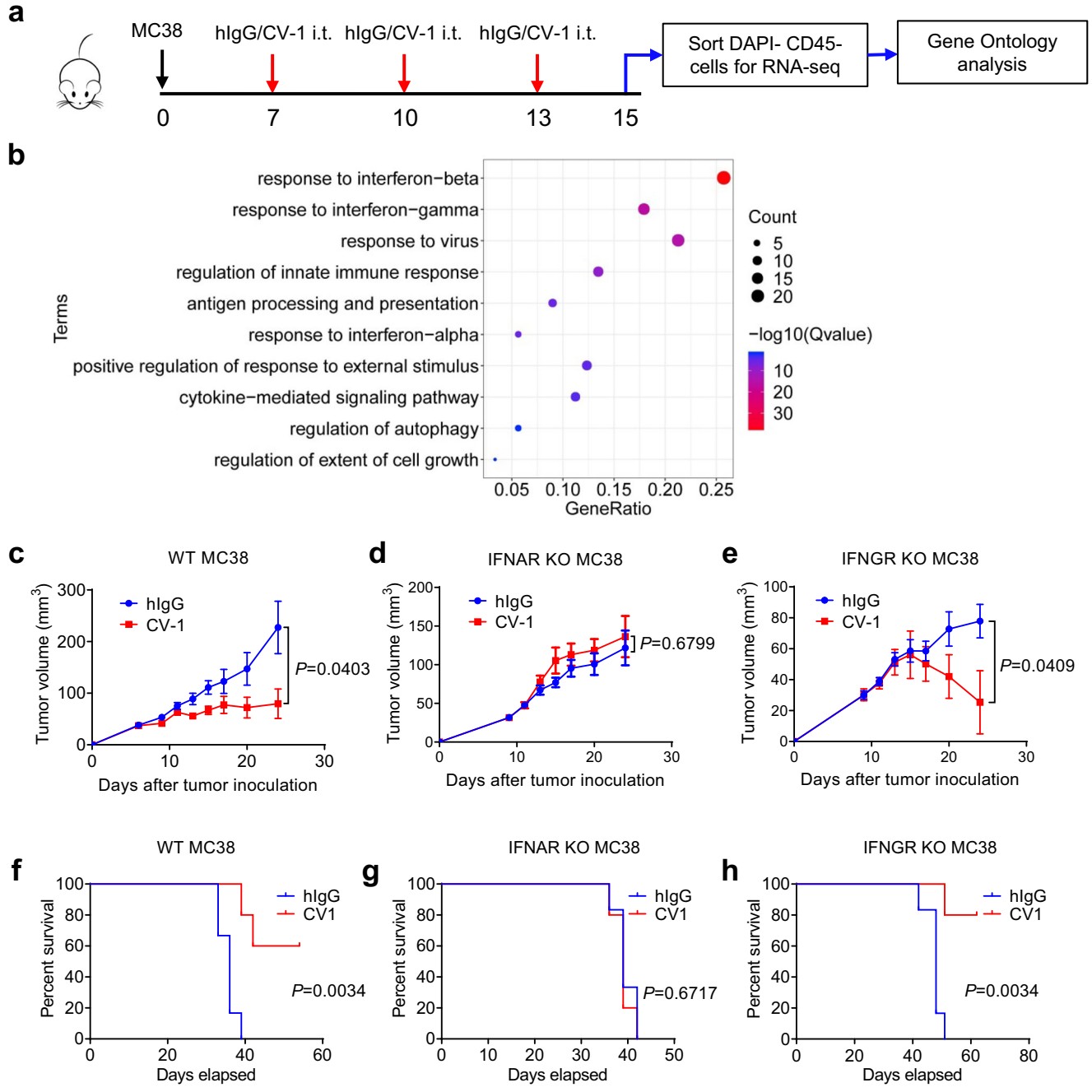

**Fig. 1 | Tumor cell-intrinsic IFN-I signaling is essential for CD47 blockade therapy in mice. a** Schematic experimental design. C57BL/6 mice bearing MC38 tumors were treated i.t. with CV-1 or hIgG every three days. Two days after the third treatment, CD45− cells in each tumor were sorted for RNA-seq. Gene Ontology analysis was performed on differentially expressed genes. (*n* = 2 biologically independent samples in hIgG group and *n* = 3 biologically independent samples in CV-1 group.) **b** Enriched pathways of genes differentially expressed on tumors treated by CV-1 compared to hIgG. **c**–**e** C57BL/6 mice (*n* = 6 mice in hIgG group and *n* = 5 mice in CV-1 group) bearing WT (**c**), IFNAR1 KO (**d**) or IFNGR1 KO (**e**) MC38 tumors were treated i.t. with CV-1 or hIgG every three days. Tumor volume was measured at indicated time. **f**–**h** The survival curves were shown corresponding to (**c**)–(**e**). Data are representative of two independent experiments in (**a**) and (**b**), three independent experiments in (**c**)–(**e**). Two-tailed unpaired Student's *t* test was used in (**c**)–(**e**). Log-rank (Mantel-Cox) test was used in (**f**)–(**h**). Data are presented as mean values ± SEM. Source data are provided as a Source Data file.

downstream of tumor cell-intrinsic IFN-I signaling-mediated antitumor effect (Fig. 2h).

## IFN-Is promote OXPHOS and ATP production in tumor cells

Next, we asked how tumor cell-intrinsic IFN-I signaling promotes ATP release. ATP can be produced efficiently via mitochondrial OXPHOS, or less efficiently via glycolysis. We first examined whether IFN-I signaling regulates mitochondrial OXPHOS. Indeed, significantly increased

mitochondrial mass (mt mass) and mitochondrial membrane potential (mtMp) were found in WT MC38 cells upon IFN-α treatment in vitro (Fig. 3a, b). The ratio of MDR to MG, an indication of mitochondrial activity per mitochondrial mass, was also decreased with IFN-α stimulation, suggesting that tumor cells might fail to fully utilize mitochondrial activity in IFN-α stimulation (Fig. 3c). Similar effects were observed in A20 and HT29 cells (Supplementary Fig. 4a–f). In addition, the copy number of mitochondrial DNA (mtDNA) in tumor cells was

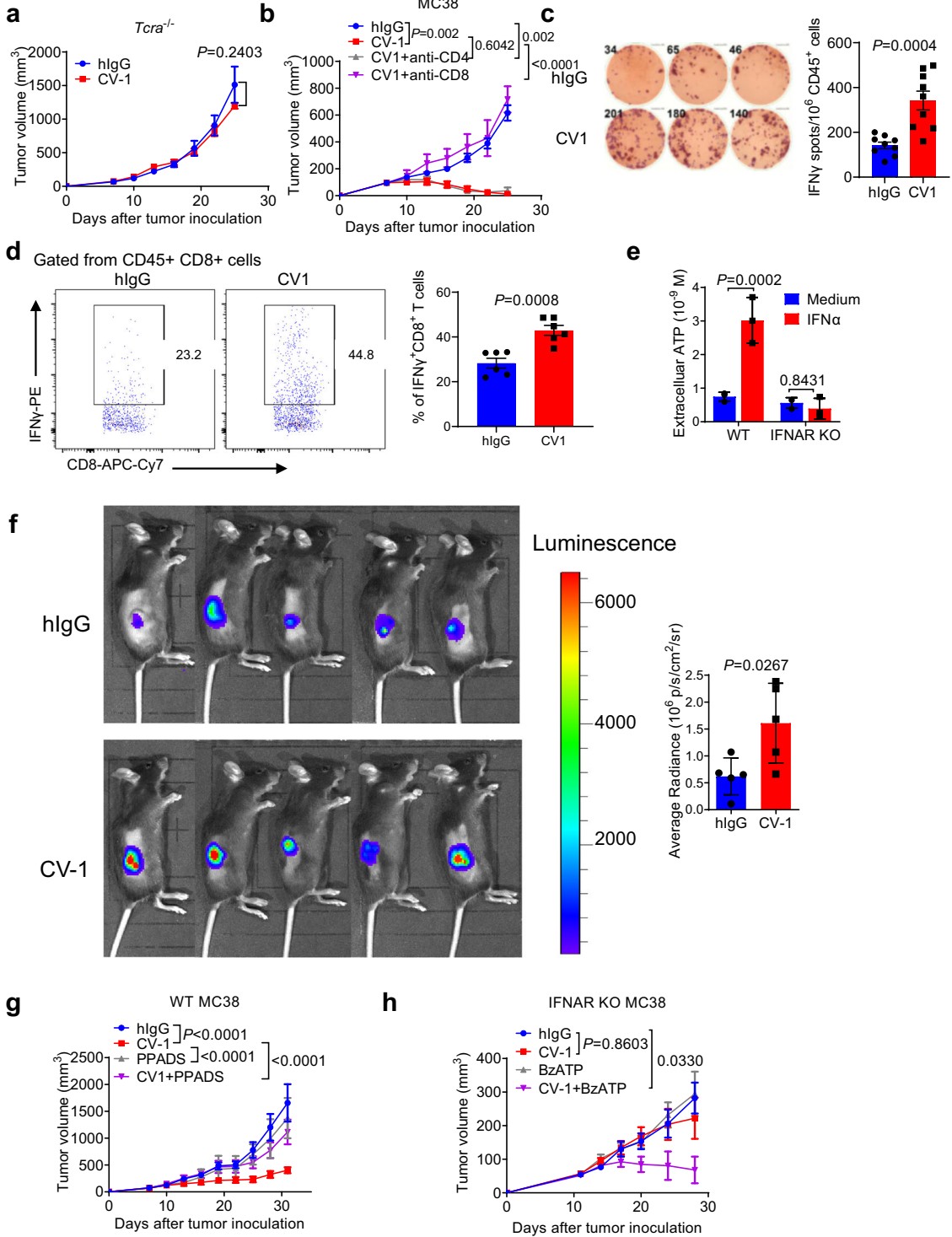

also increased upon IFN-α treatment, indicating stronger mitochondrial biogenesis (Supplementary Fig. 4g). It was reported that tumor mtDNA inside the cytosol of DCs is increased in response to CD47 blockade, which is essential for the induction of IFN-Is and antitumor immunity[55]. Thus, these results may suggest a positive feedback loop between DC IFN-I production and tumor cell mtDNA biogenesis.

Next, we directly analyzed the effect of IFN-Is on mitochondrial respiration using Seahorse assay. We observed a significant increase in basal respiration and ATP production rate in MC38 cells upon IFN-α treatment (Fig. 3d). Consistent with this result, the concentration of intracellular ATP in MC38 cells was also upregulated (Fig. 3e). Similar results were also obtained for A20 and HT29 cells (Supplementary

Fig. 4h, i). In contrast, glycolysis did not seem to be affected by IFN-α treatment within current time frame (Supplementary Fig. 4j), although the long-term effect of IFN-α on tumor cell glycolysis cannot be formally excluded. Further supporting the enhanced OXPHOS upon IFN-α treatment, we found the expression of several genes associated with OXPHOS was also upregulated (Fig. 3f), which was the same upon in vivo CV-1 treatment (Supplementary Fig. 4k). NDUFS6 is a conserved subunit of complex I and is a part of the enzymatic core of complex[56,57]. In *Ndufs6* KO MC38 cells, no increase of ATP production and release was observed upon IFN-α treatment in vitro (Fig. 3g, h). Moreover, *Ndufs6* KO MC38 tumors were resistant to CV-1 treatment in vivo (Fig. 3i). Therefore, these data suggest that IFN-Is reprogram tumor cell

**Fig. 2 | IFN-Is promote ATP release from tumor cells which is essential for CD47 blockade therapy in mice. a** Tcrα−/− mice (n = 5 mice in hIgG group and n = 6 mice in CV-1 group) bearing WT MC38 tumors were treated i.t. with CV-1 or hIgG every three days. **b** C57BL/6 mice (n = 7 mice per group) bearing WT MC38 tumors were treated i.t. with CV-1 or hIgG every three days. CD8- or CD4-depleting antibody was administered i.p. twice a week, starting on day 7. C57BL/6 mice bearing WT MC38 (**c**) or MC38-OVA (**d**) tumors were treated i.t. with CV-1 or hIgG every three days. **c** Two days after the third treatment, CD45+ cells were sorted from tumors and stimulated with MC38 tumor cell lysis, tumor-specific IFN-γ producing cells were measured by ELISPOT (n = 9 biologically independent samples). **d** Two days after the third treatment, CD45+ cells were sorted from tumors and stimulated with OT-I peptide for 6 h, with Brefeldin A added to block IFN-γ secretion. The frequencies of IFN-γ+ cells in CD8 + T cells were analyzed by FACS (n = 6 biologically independent samples per group). **e** Extracellular ATP concentration was measured from MC38 cells treated with or without IFN-α for 48 hours (n = 3 biologically independent samples per group). **f** C57BL/6 mice (n = 5 mice per group) bearing WT MC38 tumors were treated i.t. with CV-1 or hIgG every three days. Two days after the third injection, extracellular ATP concentration in tumor microenvironment was measured by IVIS. **g** C57BL/6 mice (n = 5 mice per group) bearing WT MC38 tumors were treated i.t. with hIgG, CV-1, PPADS or CV-1 plus PPADS every three days. Tumor volume was measured at indicated time. **h** C57BL/6 mice (n = 5 mice per group) bearing IFNAR1 KO MC38 tumors were treated i.t. with hIgG, CV-1, BzATP or CV-1 plus BzATP every three days. Tumor volume was measured at indicated time. Data are representative of two independent experiments in (**a**)–(**d**) and (**f**), three independent experiments in (**e**), (**g**), and (**h**). Two-tailed unpaired Student's t test was used in (**a**), (**c**), (**d**), (**f**). Two-way ANOVA and multiple comparisons test was used in (**b**), (**e**), (**g**), (**h**). Data are presented as mean values ± SEM. Source data are provided as a Source Data file.

metabolism by facilitating ATP production via mitochondrial OXPHOS, and that tumor cells with OXPHOS genetic deficiency are resistant to CD47-SIRPα blockade therapy.

## IFN-Is promote tumor cell OXPHOS and ATP production via ISG15

To further identify the mechanism by which IFN-Is promote tumor cell OXPHOS and ATP production, MC38 cells treated with IFN-α or medium in vitro were collected for RNA-seq analysis. *Isg15*, an IFN α/β-induced gene encoding a ubiquitin-like protein, was found significantly upregulated at 6 h, 24 h and 48 h (Fig. 4a, and Supplementary Fig. 5a). CV-1 in vivo treatment also significantly upregulated *Isg15* gene expression in MC38 cells (Fig. 4b). ISG15 has been well-documented to govern mitochondrial function in several cell types including tumor cells, mainly via ISGylation of mitochondrial proteins[58–60]. Therefore, we tested its role in tumor cells. *Isg15* KO MC38 cell lines were constructed using CRISPR-Cas9 (Supplementary Fig. 5b). In *Isg15* KO MC38 cells, no increase of ATP production or release was observed upon IFN-α treatment (Fig. 4c, d and Supplementary Fig. 5c). Seahorse assay also showed no increase in basal respiration and ATP production in *Isg15* KO MC38 cells upon IFN-α treatment (Fig. 4e and Supplementary Fig. 5d). In vivo, *Isg15* deficiency in MC38 tumor cells almost completely abolished the therapeutic effect of CV-1 (Fig. 4f and Supplementary Fig. 5e). These results suggest that IFN-I-induced ISG15 is a major factor promoting OXPHOS and ATP production in tumor cells, and ISG15 is crucial for the efficacy of CD47-SIRPα blockade therapy.

## IFN-Is induce autophagy in tumor cells which is essential for ATP release

Extracellular ATP release is actively regulated via autophagy upon chemical drug treatment[61]; IFN-I has been reported to promote autophagy in several human cancer types[62]. Consistent with the latter, an enriched autophagy pathway was also found in GO analysis, as shown in Fig. 1b. Therefore, we explored whether IFN-Is might facilitate extracellular ATP release via autophagy. We first confirmed that IFN-α treatment increased the formation of autophagosomes and the LC3-II/LC3-I ratio in MC38 cells (Fig. 5a, b), which was also true in HT29 cells (Supplementary Fig. 6a). In addition, autophagy-related genes, including *Nupr1*, *Trib3*, *Trim21* and *Tmem150a* were upregulated following IFN-α stimulation in vitro and CV-1 treatment in vivo (Supplementary Fig. 6b, c). Second, using quinacrine to visualize ATP-containing vesicles, we observed that IFN-Is promoted the enrichment of ATP into vesicles in WT MC38 cells (Fig. 5c) and HT29 cells (Supplementary Fig. 6d). As a control, *Atg5* deficient MC38 cells showed insignificant change of ATP-containing vesicles upon IFN-α treatment (Fig. 5c, Supplementary Fig. 6e). Moreover, *Atg5* deficiency largely abolished extracellular ATP release upon IFN-α treatment (Fig. 5d), suggesting a critical role of autophagy on IFN-α induced ATP release. It should be noted that autophagy may be also required for intracellular ATP production to some degree (Supplementary Fig. 6f)[63],

although this effect was not complete. The exact mechanism of autophagy in ATP production remain to be investigated in future.

Autophagy in tumor cells plays a multifaceted role in tumor development, progression and therapy[64–66]. Given its essential role in IFN-I-induced extracellular ATP release, we explored its role in CD47 blockade therapy. *Atg5* deficiency in MC38 tumor cells almost completely abolished the therapeutic effect of CV-1 (Fig. 5e). Together, these results indicate that IFN-Is promote the extracellular release of ATP in an autophagy-dependent manner and that autophagy is essential for the therapeutic effect of CD47-SIRPα blockade.

## IFN-Is do not increase ATP production and release in myeloid cells

To evaluate the possibility that IFN-I may also regulate ATP production and release in myeloid cells that are abundantly present in TME, we first examined ATP production and release in LPS-primed bone marrow-derived macrophages (BMDMs) after IFN-α treatment. In contrast to those found in MC38 cells, IFN-α did not promote, but even inhibited, the intracellular and extracellular ATP concentrations in BMDMs (Fig. 6a, b). Consistently, BMDMs did not show increased autophagy after IFN-α treatment (Fig. 6c). Furthermore, in vivo CV-1 treatment significantly increased LC3-II/LC-3-I ratio in tumor cells but not in CD45+ hematopoietic cells (Fig. 6d). These results suggest tumor cells but not hematopoietic cells in TME are the dominant responder to IFN-Is for ATP production and extracellular release.

## IFN-γ does not upregulate tumor cell ATP production and release

Given the shared pathway between IFN-Is and IFN-II, we next investigated the role of IFN-II, specifically IFN-γ, in ATP production and release in tumor cells. Interestingly, in contrast to IFN-Is, IFN-γ treatment did not alter intracellular ATP production or extracellular ATP release (Fig. 7a, b). Although IFN-γ treatment also upregulated *Isg15* expression in MC38 to some degree, the effect was far less than IFN-α treatment (Fig. 7c). Quantitative real-time PCR analysis also confirmed that IFN-γ had no obvious effect on the expression of genes related to OXPHOS or autophagy, in stark contrast to the effect of IFN-α (Fig. 7d, e). LC3-II/LC3-I ratio was also not changed in MC38 cells upon IFN-γ treatment (Fig. 7f). The activity of IFN-γ is intact, since *Irf8* and *Ido1*, typical IFN-γ-specific interferon-stimulated genes (ISGs), were both found to be dramatically upregulated by IFN-γ, but not IFN-α (Fig. 7g). Thus, the effect on ATP production and release is specific to IFN-Is but not IFN-γ in tumor cells.

## Tumor cell-intrinsic IFN-I signaling is required for T cell response induction

The previous results showed increased number of tumor infiltrating tumor antigen-specific CD8 + T cells upon CV-1 treatment and CD8 + T cell is critical for CV-1 efficacy (Fig. 2a–d). However, at which stage of T cell response, namely induction phase or effector phase, tumor cell

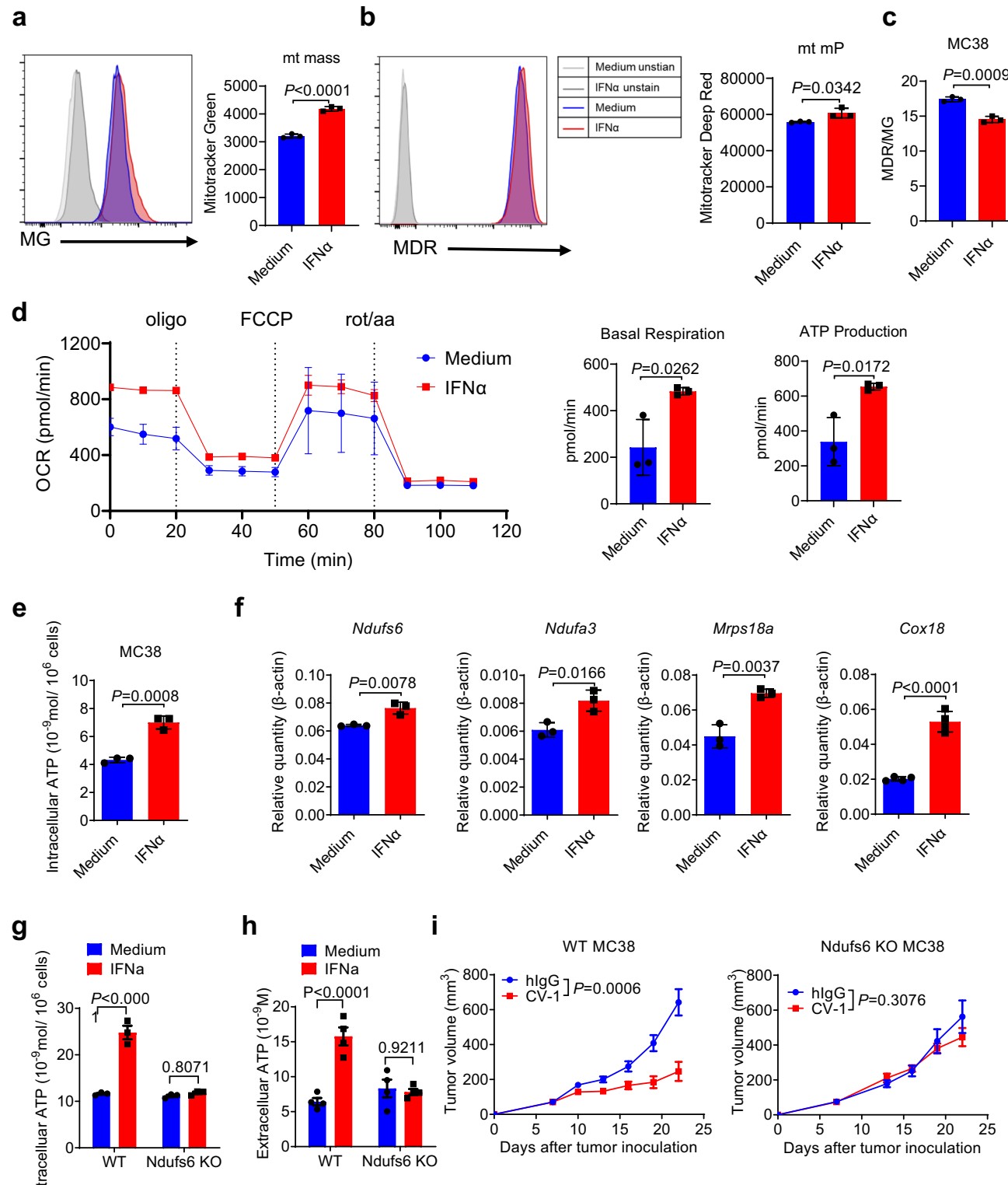

**Fig. 3 | IFN-Is promote OXPHOS and ATP production in tumor cells. a, b** WT MC38 cells were treated with IFN-α for 48 hours. Mitochondrial mass (**a**) and membrane potential (**b**) were measured by FACS ($n = 3$ biologically independent samples per group). **c** The ratio of MDR to MG was shown. **d** OCR was measured by Seahorse assay ($n = 3$ biologically independent samples per group). **e** Intracellular ATP concentration was measured ($n = 3$ biologically independent samples per group). **f** The expression of OXPHOS related genes were measured by real-time PCR ($n = 3$ biologically independent samples per group). **g, h** Ndufs6 KO MC38 cells were treated with IFN-α for 48 hours. Concentration of intracellular ATP ($n = 3$ biologically independent samples per group) (**g**) and extracellular ATP ($n = 4$ biologically independent samples per group) (**h**) were measured. **i** C57BL/6 mice ($n = 5$ mice per group) bearing WT or Ndufs6 KO MC38 tumors were treated i.t. with CV-1 or hIgG every three days. Tumor volume was measured at indicated time. Data are representative of four independent experiments in (**a**)–(**c**), (**e**), (**g**), and (**h**), three independent experiments in (**d**), two independent experiments in (**f**), **i** Two-tailed unpaired Student's *t* test was used in (**a**)–(**f**). **i** Two-way ANOVA and multiple comparisons test was used in (**g**) and (**h**). Data are presented as mean values ± SEM. Source data are provided as a Source Data file.

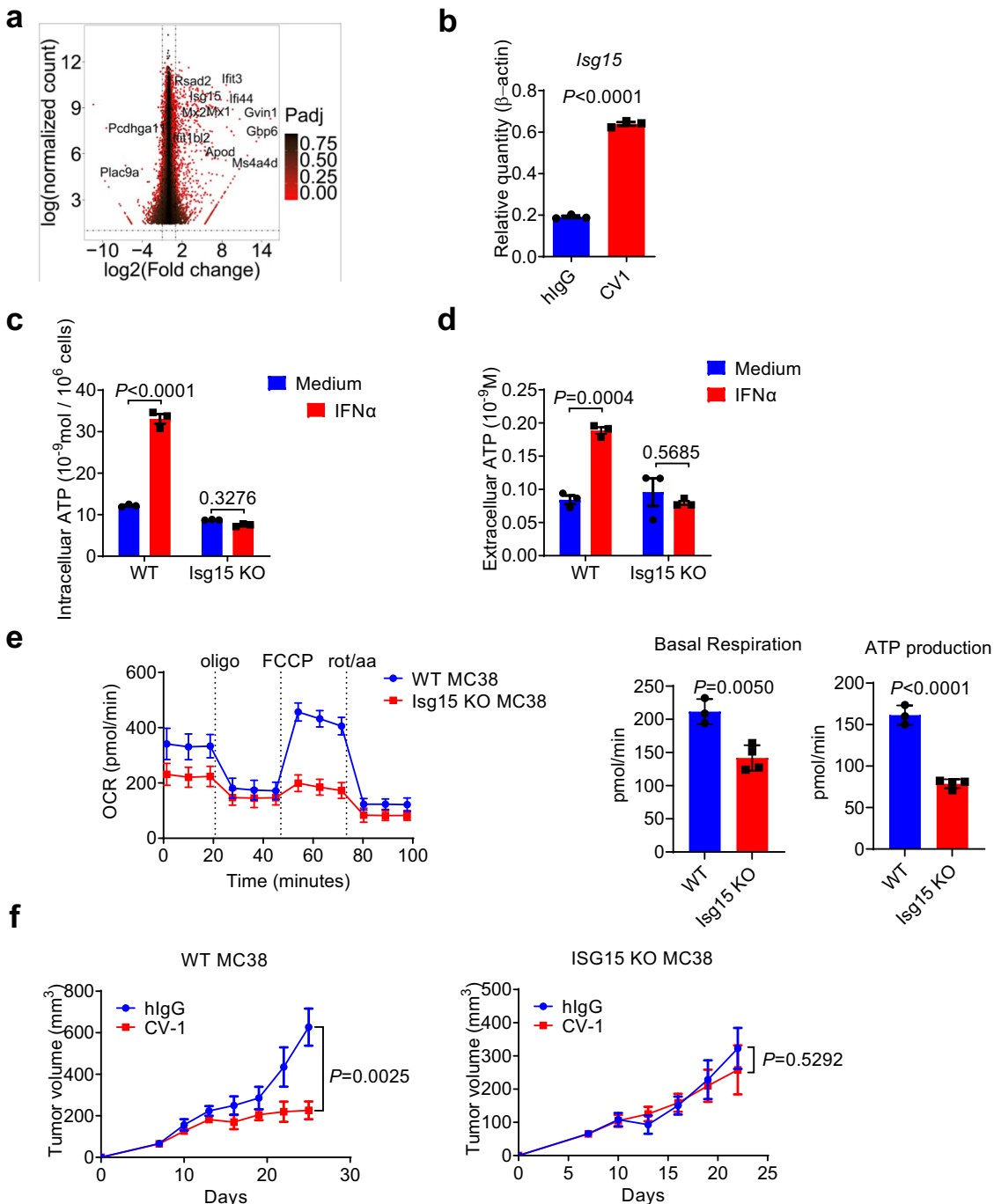

**Fig. 4 | IFN-Is promote tumor cell OXPHOS and ATP production via ISG15.**
**a** RNA-seq analysis of WT MC38 cells stimulated with IFN-α for 48 h. The differentially expressed genes between the medium and IFN-α treatment groups are shown in a volcano plot. ($n = 3$ biologically independent samples per group.)
**b** C57BL/6 mice bearing WT MC38 tumors were treated i.t. with CV-1 or hIgG every three days. Two days after the third treatment, CD45− cells were sorted. The expression of *Isg15* gene were measured by real-time PCR ($n = 3$ biologically independent samples per group). **c, d** WT and Isg15 KO MC38 cells were treated with IFN-α for 24 h. The concentration of intracellular ATP (**c**) and extracellular ATP (**d**) were measured ($n = 3$ biologically independent samples per group). **e** OCR was measured using Seahorse assay ($n = 3$ biologically independent samples in WT group and $n = 4$ biologically independent samples in Isg15 KO group). **f** C57BL/6 mice bearing WT ($n = 6$ mice per group) or Isg15 KO MC38 ($n = 4$ mice per group) tumors were treated with CV-1 or hIgG every three days. The tumor volume was measured at the indicated time. Data are representative of two independent experiments in (**a**) and (**f**), three independent experiments in (**b**)–(**e**). DESeq2 package, with Negative Binomial GLM fitting and Wald statistics were used in (**a**). Two-tailed unpaired Student's *t* test were used in (**b**), (**e**), and (**f**). Two-way ANOVA and multiple comparisons test was used in (**c**) and (**d**). Data are presented as mean values ± SEM. Source data are provided as a Source Data file.

IFN-I signaling is required is unknown. To clarify this, we adopted a delicate model in which mice were inoculated with WT MC38 cells on one flank and IFNAR1 KO cells on the other[67] (Fig. 8a). CV-1 was injected only into the right-side (WT or IFNAR1 KO) tumors. We then measured the growth of both injected and distant (non-injected) tumors. When WT tumors were injected with CV-1, both WT and distant IFNAR1 KO tumors were inhibited (Fig. 8a). In contrast, when IFNAR1 KO tumors were injected with CV-1, neither IFNAR1 KO nor distant WT tumors were inhibited (Fig. 8a). When the tumor infiltrating CD8+ T cells were detected, we found that CV-1 injection into WT tumors increased the

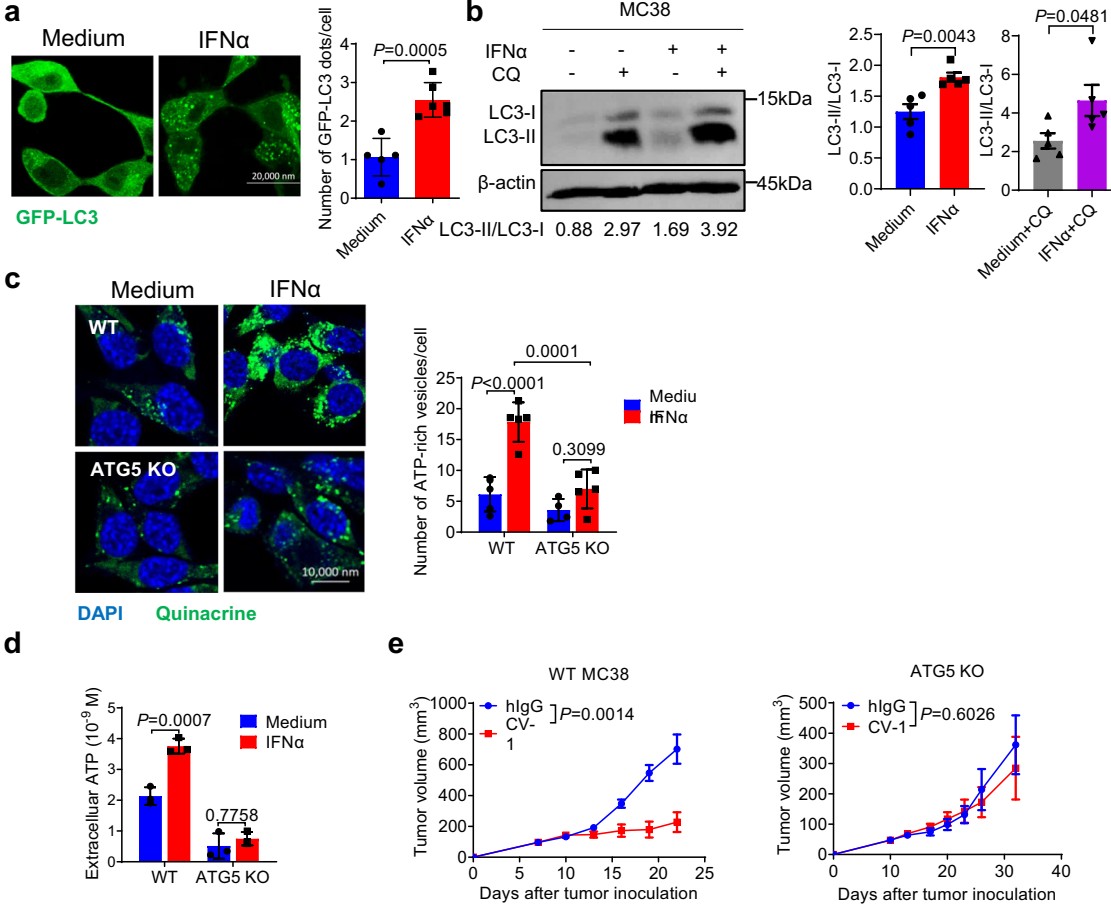

**Fig. 5 | IFN-Is induce autophagy in tumor cells for ATP extracellular release.**
**a** MC38-GFP-LC3 cells were treated with IFN-α for 48 hours. Autophagosomes were measured by immunofluorescence ($n = 5$ biologically independent samples in Medium group and $n = 6$ biologically independent samples in IFN-α group). **b** MC38 cells were treated with IFN-α for 48 hours with or without chloroquine (CQ). Autophagy was measured by Western blot ($n = 3$ biologically independent samples). WT and Atg5 KO MC38 cells were treated with IFN-α for 48 hours. **c** ATP-rich vesicles were measured by immunofluorescence ($n = 5$ biologically independent samples). **d** Extracellular ATP concentration was measured ($n = 3$ biologically

independent samples). **e** C57BL/6 mice ($n = 5$ mice per group) bearing WT or ATG5 KO MC38 tumors were treated i.t. with CV-1 or hIgG every three days. Tumor volume was measured at indicated time. Data are representative of three independent experiments in (**a**), (**c**), and (**d**), five independent experiments in (**b**), two independent experiments in (**e**). Two-tailed unpaired Student's *t* test was used in (**a**), (**b**), and (**e**). Two-way ANOVA and multiple comparisons test were used in (**c**) and (**d**). Data are presented as mean values ± SEM. Source data are provided as a Source Data file.

percentage of CD8+ T cells among total cells in both injected WT and distant IFNAR1 KO tumors. However, when IFNAR1 KO tumors were injected, the percentage of CD8+ T cells in both injected IFNAR1 KO and distant WT tumors remained unchanged compared to the control treatment (Fig. 8b). Thus, these data suggest that CV-1 treatment induces a systemic antitumor response in the presence of tumor cell-intrinsic IFN-I signaling, and this response is effective on IFNAR1 KO tumors; IFNAR1 deficiency in tumor cells does not influence the effector T cell killing function at the effector phase. It should be noted that only the percentage of CD8 + T cell was measured in these assays. For more detailed evaluation, the absolute CD8 + T cell count per tumor weight and the activation status of these infiltrating CD8 + T cells should be measured.

We then asked how extracellular ATP is involved in T cell response induction. It has been reported that extracellular ATP stimulates DC activation and promotes the release of IL-1β to activate CD8 + T cells[68]. To determine whether IFN-I signaling in tumor cells affects DC activation, CD80 expression was measured. Upon CV-1 treatment, CD80 expression was significantly upregulated in DCs isolated from WT MC38 tumor-bearing mice, but not from IFNAR1 KO MC38 tumor-bearing mice (Fig. 8c, Supplementary Fig. 7a). Furthermore, DCs isolated from CV-1-treated MC38-OVA tumors showed a slightly but

significantly enhanced ability to activate naïve OT-I T cells than those isolated from control hIgG treated tumors (Fig. 8d), suggesting an enhanced DC function for T cell activation upon CV-1 treatment. It should be noted that the low percentage of T cell activation is probably due to the limitation of the assay. MC38-OVA tumor model was used here, in which tumor cell-endogenous OVA antigen expression, release, capture, processing and presentation by tumor-associated DCs might not be so prevalent that can be easily detected.

To directly evaluate the role of DC responsiveness to extracellular ATP in CV-1 treatment, we generated a mouse model in which P2X7R is specifically deficient in DCs but not other cells. Notably, P2X7R deficiency in DCs led to almost complete resistance of tumors to CV-1 treatment, indicating a crucial role of extracellular ATP-mediated DC activation in this treatment (Fig. 8e). To further evaluate the effect of IL-1β in CV-1 treatment, IL-1Ra, an IL-1β antagonist, was prepared (Supplementary Fig. 7b, c). IL-1β blockade also abolished the therapeutic effect of CV-1 (Fig. 8f). Together, these data suggest that tumor cell-intrinsic IFN-I signaling is essential for anti-CD8+ T cell response induction likely via DC activation and IL-1β production in the context of CD47-SIRPα blockade therapy. It should be noted that the current data do not exclude the possibility that effector T cell activation and maintenance in the tumors might

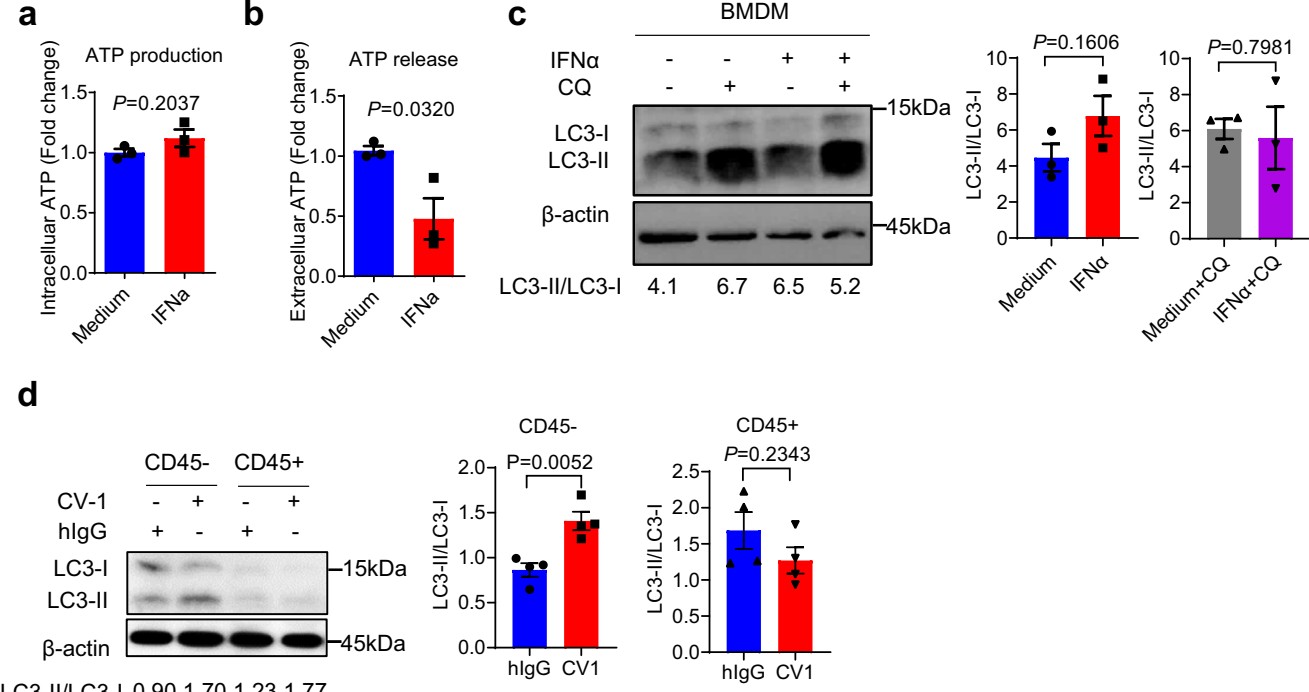

**Fig. 6 | IFN-Is do not increase ATP production and release in myeloid cells.**
**a**, **b** BMDM were treated with IFN-α for 48 h ($n = 3$ biologically independent samples per group). The concentrations of intracellular ATP (**a**) and extracellular ATP (**b**) were measured. **c** BMDM cells were treated with IFN-α for 48 h with or without CQ. Autophagy was measured by Western blot ($n = 3$ biologically independent samples per group). **d** C57BL/6 mice bearing WT MC38 tumors were treated i.t. with CV-1 or hIgG every three days. Two days after the third injection, CD45− and CD45+ cells were sorted for autophagy detection ($n = 4$ biologically independent samples per group). Data are representative of three independent experiments in (**a**)−(**c**), four independent experiments in (**d**). Two-tailed unpaired Student's $t$ test were used. Data are presented as mean values ± SEM. Source data are provided as a Source Data file.

be also increased due to enhanced DC activation. Further investigation is required.

## CD47 blockade synergizes with CD39 or CD73 inhibition

While extracellular ATP has stimulatory effects for antitumor T cell response, it is easily degraded to suppressive adenosine given the prevalent expression of CD39 and CD73 in TME[69–71]. We confirmed the expression of CD39 on Treg cells as well as tumor infiltrating CD8+ T cells and the expression of CD73 mainly in tumor cells (Fig. 9a, b, Supplementary Fig. 8a). The expressions of CD39 and CD73 were not affected by CV-1 treatment (Supplementary Fig. 8b, c). To test whether CD39/CD73 may dampen the antitumor effect of CV-1 treatment, we first constructed a CD39-overexpression MC38 cell line (Fig. 9c). Results showed that overexpression of CD39 in tumor cells totally abolished the therapeutic effect of CV-1 (Fig. 9d). To further test whether inhibition of ATP degradation might enhance the therapeutic efficacy of CV-1, we treated the WT MC38 tumors with CV-1 together with a CD39 inhibitor, POM-1. Indeed, the combination treatment showed significantly increased efficacy than single CV-1 or POM-1 treatment (Fig. 9e). In addition, the combination of CV-1 and anti-CD73 in WT MC38 tumors also significantly enhanced tumor growth inhibition (Fig. 9f). Thus, these data suggest that inhibition of ATP degradation, via CD39 or CD73 inhibition, may further enhance the antitumor immunity induced by CD47-SIRPα blockade.

## Discussion

ICB has revolutionized the field of cancer therapy in the past decade. However, approximately two-thirds of patients or more still suffer from therapeutic resistance[72]. This is especially true for CD47-SIRPα ICB in solid tumors[7,14,15]. Identifying tumor cell-intrinsic factors controlling CD47-SIRPα blockade resistance is an urgent task in this field[73]. Here, using several murine tumor models, including colorectal cancer

and B cell lymphoma, we revealed that tumor responsiveness to IFN-Is, but not IFN-II, is critical for the efficacy of CD47-SIRPα ICB. Tumor cell-intrinsic OXPHOS and autophagy are both activated by IFN-Is, leading to increased ATP production and extracellular release, which induces antitumor T cell response (Supplementary Fig. 9). The effect of IFN-I on tumor cell OXPHOS and autophagy, and consequent ATP production and release, were also confirmed in HT29 human tumor model. In addition, combination therapy with CD39/CD73 inhibitors further improves the efficacy of CD47-SIRPα ICB. Thus, our study revealed several tumor cell-intrinsic mechanisms underlying the resistance/efficacy of CD47-SIRPα ICB, and provided strategies to overcome resistance or enhance efficacy, shedding light on the clinical development of CD47-SIRPα ICB.

Our study suggests that overlooked tumor OXPHOS can be harnessed for enhanced tumor immunotherapy. Tumor cells are not merely the target of tumor immunity but also shape the immune community of TME. Aerobic glycolysis is one of the most common features of tumor cells. Extensive studies have demonstrated that aerobic glycolysis broadly influences antitumor immunity via lactic acid accumulation and the resulting acidic TME[27–34]. In contrast, the role of OXPHOS, a dysregulated metabolism in tumor cells, in tumor immunity and therapeutic efficacy remains unclear. In the current study, we found that tumor cell OXPHOS product ATP plays a critical role during CD47-SIRPα ICB. ATP inhibition significantly reduced the therapeutic efficacy of CD47-SIRPα ICB, and ATP administration largely rescued the failed efficacy of CD47-SIRPα ICB in IFNAR1 knockout tumors, suggesting that ATP is a major mediator downstream of tumor cell-intrinsic IFNAR signaling. Thus, proper reprogramming of tumor metabolic state from aerobic glycolysis to OXPHOS may be employed in future immunotherapy. It is important to note that OXPHOS in most cancer cells is not permanently aberrant but rather occurs at lower rates, given the limited oxygen supply in the hypoxic tumor

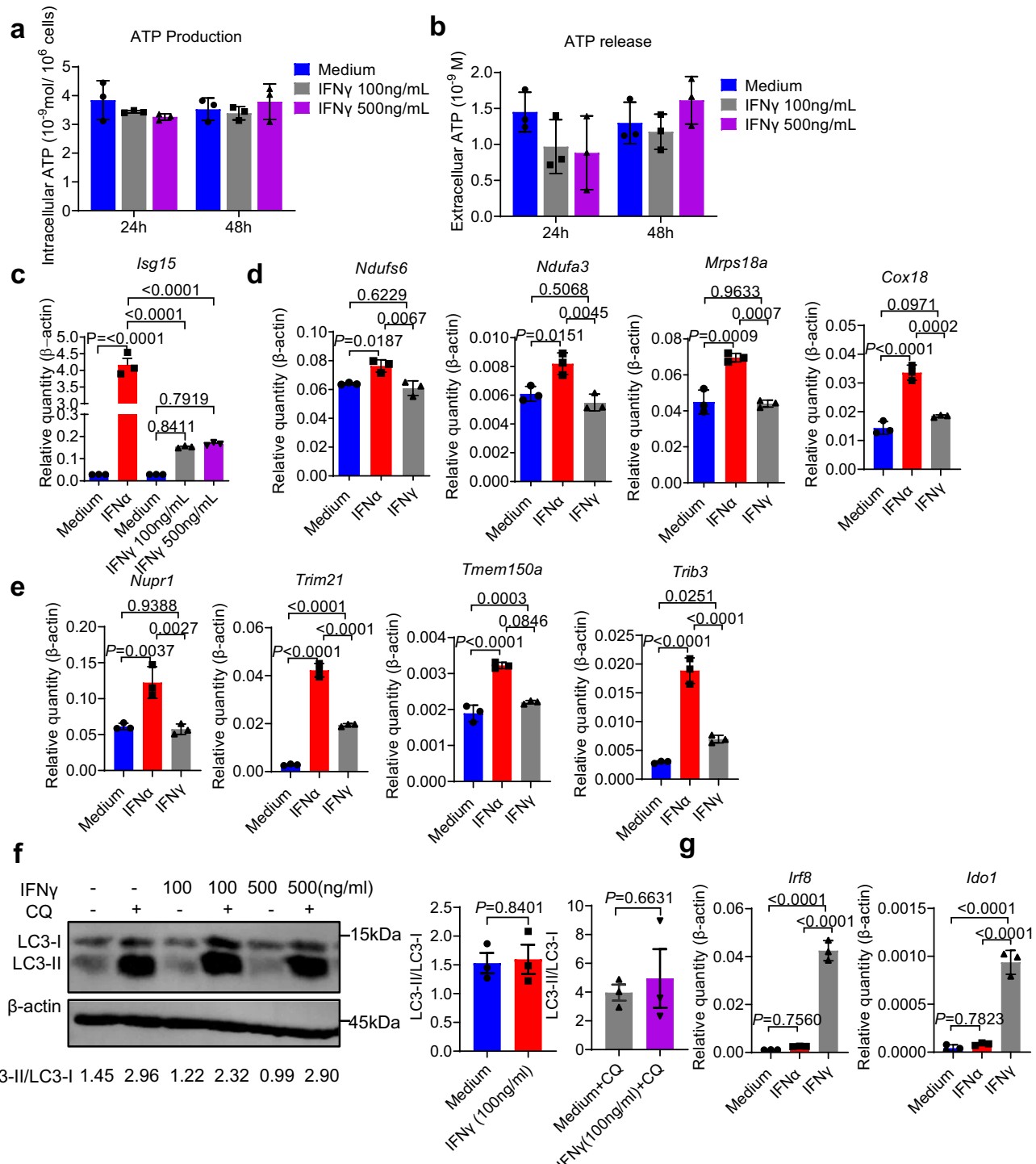

**Fig. 7 | IFN-II does not regulate tumor cell ATP production and release. a**, **b** WT MC38 cells were treated with IFN-γ for 24 h and 48 h. Concentrations of intracellular ATP (**a**) and extracellular ATP (**b**) were measured (*n* = 3 biologically independent samples per group). MC38 cells were treated with IFN-α or IFN-γ for 48 h. **c**–**e** The expression of *Isg15* (**c**) OXPHOS (**d**) and autophagy (**e**) regulating genes was measured by quantitative real-time PCR (*n* = 3 biologically independent samples per group). **f** MC38 cells were treated with IFN-γ for 48 h. Autophagy was measured by Western blot (*n* = 3 biologically independent samples per group). **g** The expression

of *Irf8* and *Ido1* was measured by quantitative real-time PCR (*n* = 3 biologically independent samples per group). Data are representative of three independent experiments in (**a**), (**b**), and (**f**), two independent experiments in (**c**)–(**e**) and (**g**). Two-tailed unpaired Student's *t* test were used in (**f**). Two-way ANOVA and multiple comparisons test was used in (**a**) and (**b**). One-way ANOVA and multiple comparisons test was used in (**c**)–(**e**) and (**g**). Data are presented as mean values ± SEM. Source data are provided as a Source Data file.

microenvironment. This further emphasizes the possibility of regulating this metabolic pathway for enhanced efficacy.

In our study, eATP effect on DCs and consequent T cell response induction appears an essential mechanism. DCs isolated from tumors

treated with CV-1 showed significantly higher level of activation and primed antigen-specific T cells more potently than DCs isolated from control treated tumors. Conditional deletion of P2X7 receptor from DCs almost completely abolished the efficacy of CD47-SIRPα ICB.

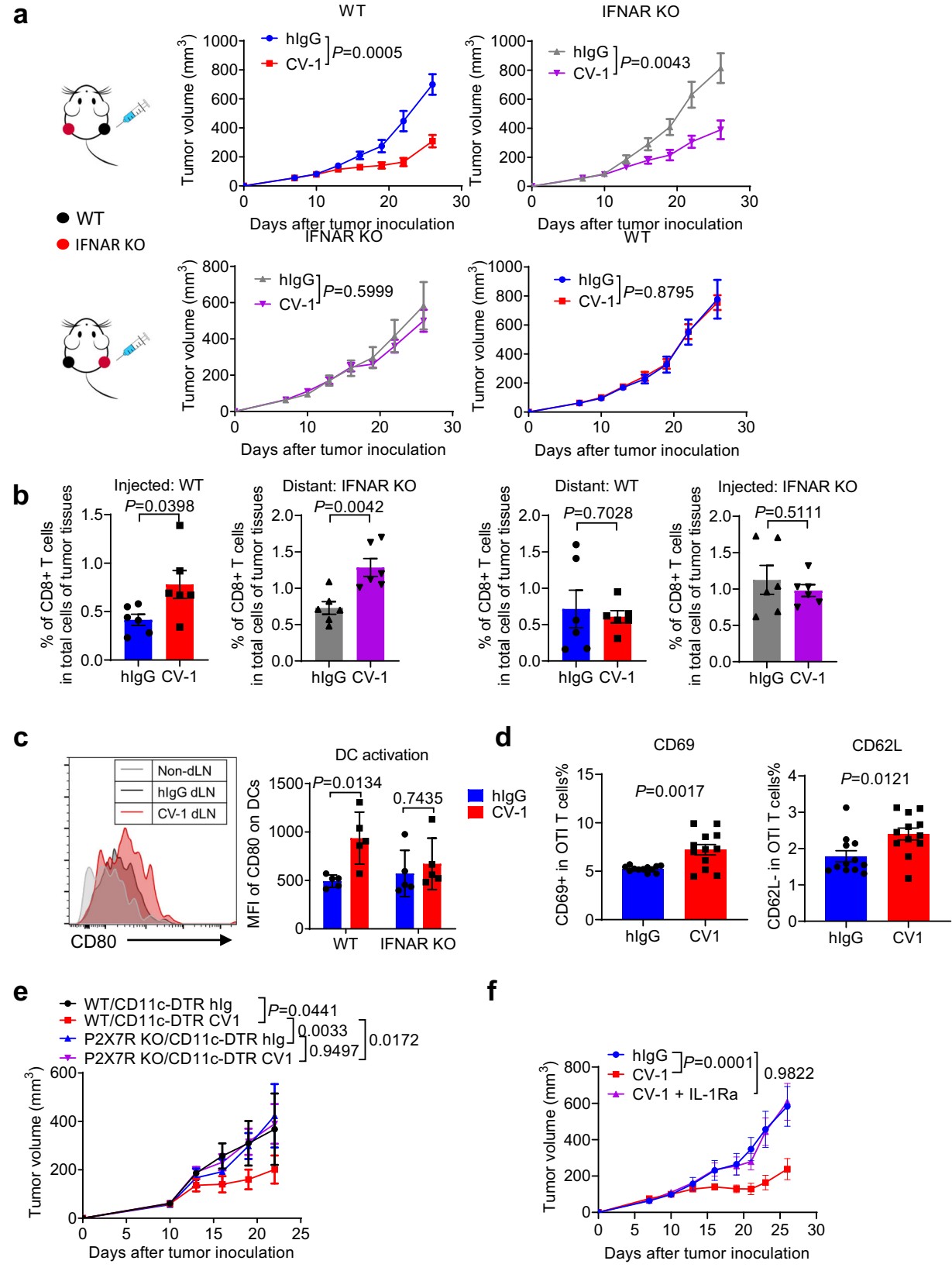

These data suggest that DCs are important responders of eATP. However, our data do not exclude the possibility that eATP may also work through other cells, such as macrophages, T cells or even tumor cell themselves for better tumor growth control. These mechanisms are not necessarily mutually exclusive to eATP's effect on DCs, further investigation is required.

In addition to ATP, ROS is an important byproduct of OXPHOS. ROS has complicated effects on tumor cells. At low to moderate concentrations, ROS directly promotes activation of multiple pathways leading to tumor cell proliferation, including CDK2[74], HIF1-α[75], PI3K/AKT/mTOR[76–78], and MAPK/ERK[79–81]. On the opposite, exceeding ROS promotes tumor cell programed cell death, via apoptosis[82,83],

**Fig. 8 | Tumor cell-intrinsic IFN-I signaling is required for T cell response induction. a** C57BL/6 mice were inoculated with WT MC38 cells on the right side and IFNAR1 KO MC38 cells on the left side (*n* = 7 mice per group), or IFNAR1 KO MC38 cells on the right side and WT MC38 cells on the left side (*n* = 7 mice in hIgG group and *n* = 6 mice in CV-1 group). Mice were treated i.t. with CV-1 or hIgG every three days for four times on the right side. Tumor growth was measured at indicated time. **b** Ten days after the fourth injection, tumors were isolated and digested. The percentage of tumor-infiltrated CD8⁺ T cells were measured (*n* = 6 mice per group). **c** C57BL/6 mice bearing WT MC38 tumors were treated i.t. with CV-1 or hIgG (*n* = 5 mice per group) every three days. Two days after the third injection, draining lymph nodes (dLNs) were isolated. The expression of CD80 on DCs in the dLNs was shown. **d** C57BL/6 mice bearing MC38-OVA tumors were treated i.t. with CV-1 or hIgG every three days. Two days after the third injection, DCs were sorted from tumors and incubated with OT-I T cells in 1:10 ratio. The expressions of CD69 and CD62L on OT-I T cells was detected 24 h after incubation by FACS (*n* = 12 biologically independent samples per group). **e** WT/CD11c-DTR and P2X7R KO/CD11c-DTR mixed bone marrow chimeras (*n* = 6 mice per group) bearing WT MC38 tumors were treated i.t. with CV-1 or hIgG every three days. Tumor volume was measured at indicated time. **f** C57BL/6 mice (*n* = 6 mice per group) bearing WT MC38 tumors were treated i.t. with hIgG, CV-1, IL-1Ra or CV-1 plus IL-1Ra every three days. Tumor volume was measured at indicated time. Data are representative of three independent experiments in (**a**)–(**c**) and (**e**), two independent experiments in (**d**) and (**f**). Two-tailed unpaired Student's *t* test was used in (**a**), (**b**), and (**d**). Two-way ANOVA and multiple comparisons test was used in (**c**), (**e**), and (**f**). Data are presented as mean values ± SEM. Source data are provided as a Source Data file.

necroptosis[84], and ferroptosis[85]. In addition, ROS may influence tumor growth indirectly via immune cells[37,38]. The direct effect of ROS on tumor cell proliferation or death unlikely contributes to tumor growth inhibition in the CD47-SIRPα blockade scenario, since without T cell-mediated immune response, the blockade cannot inhibit tumor growth at all (Fig. 2a, b). However, whether ROS-mediated tumor cell death may activate immune cells for an enhanced antitumor response remains to be studied in future.

Although IFN-II signaling pathway shares similar signaling components and some common target genes with IFN-I pathway, it has no role on tumor cell OXPHOS or autophagy. Consistently, IFN-II signaling has no or only minor stimulating effect on those genes associated with OXPHOS or autophagy. Consequently, IFNGR1 deficiency in MC38 tumor cells did not influence the efficacy of CD47-SIRPα ICB. Similar results were also found in CT26 and A20 tumors, suggesting a prevalence mechanism. Interestingly, however, the IFNGR1 signaling pathway in tumor cells has recently been shown to be essential for the efficacy of CTLA-4 and PD-1 ICB. Melanoma tumors with loss of IFNGR signaling are resistant to CTLA-4 ICB therapy, probably due to the attenuated suppression of tumor cell proliferation and apoptosis[86]. In a recent CIRSPR screen, defects in the IFNGR pathway (*Stat1, Jak1, Ifngr2, Ifngr1* and *Jak2*) were found to induce resistance to PD-1 ICB, associated with failed upregulation of MHC-I and decreased sensitivity to CTL[87]. Clinical studies also confirmed these findings[88]. The differential requirement of tumor cell IFNGR signaling for CTLA-4 or PD-1 ICB and IFNAR signaling for CD47-SIRPα ICB may be due to the difference of their primary effector targets. While CTLA-4 or PD-1 ICB targets T cells, at which phase IFN-γ is a major effector cytokine, CD47-SIRPα ICB targets APCs, at which phase IFN-Is are probably more critical. In future, the differential requirement of IFN pathway in innate ICB compared to adaptive ICB probably should be considered for precision treatment.

In summary, the highly varied genetic, epigenetic, and metabolic features of tumors are major reasons for the varied outcomes of immunotherapies. Our study highlights that tumor cell IFN-I responsiveness and the downstream OXPHOS and autophagy pathways are highly required for CD47-SIRPα ICB. Properly employing these mechanisms, such as inhibiting ATP degradation, can significantly enhance the efficacy of CD47-SIRPα ICB. This might open a window in the current dilemma of unsatisfactory efficacy of CD47-SIRPα ICB, particularly for solid tumors.

## Methods

### Mice
WT C57BL/6 or BALB/c mice were purchased from Vital River, a Charles River company in China. P2X7R KO mice were provided by Prof. Jinhui Tao (University of Science and Technology of China, Hefei, China). *Tcra*⁻/⁻ and OT-I TCR transgenic mice were purchased from The Jackson Laboratory. All mice were maintained under specific pathogen-free conditions and all animal experimental procedures were performed with approval (SYXK2020035) from the institutional committee of the Institute of Biophysics, Chinese Academy of Sciences. The housing conditions for the mice were 12-hour light/12-hour dark cycle, temperatures of 20-26 °C with 40-70% humidity. The sex of animals used in the study was female, and the age was 7–8 weeks.

### Cells and regents
MC38 cells are a murine colon adenocarcinoma cell line derived from C57BL/6 mice. A20 cells are a murine lymphoma cell line derived from BALB/c mice. CT26 cells are a murine colorectal carcinoma cell line derived from BALB/c mice. HT29 cells are a human colon cancer cell line. MC38 and CT26 cells were maintained in DMEM (Invitrogen) supplemented with 10% FBS and 1% penicillin-streptomycin. A20 and HT29 cells were cultured in RPMI 1640 (Invitrogen) supplemented with 2 mM L-glutamine, 1 mM sodium pyruvate, 0.1 mM nonessential amino acid, 1% penicillin-streptomycin, 2-ME, and 10% FBS. Recombinant proteins CV-1, Miap301, and IFN-α were kindly provided by Dr. Yang-Xin Fu. IFN-γ was purchased from GenScript (Z02916, Beijing, China). Anti-CD73 antibody was purchased from BioXCell (BE0209). BzATP was purchased from Sigma (112898-15-4). PPADS was purchased from Abcam (ab120009; Cambridge, UK). POM-1 was purchased from the Cayman Chemical Co. (21160). IFN-α2b was purchased from GenScript (Z03002, Beijing, China).

### CRISPR/Cas9 and gene overexpression
IFNAR1, IFNGR1, Atg5, Ndufs6, and Isg15 deficient tumor cell lines were generated using CRISPR/Cas9 technology. Guide RNA was designed at https://zlab.bio/guide-design-resources and the sequences are listed in Supplementary Table 1. Monoclones were screened and confirmed by DNA sequencing, FACS staining or Western blot.

For CD39 overexpression, 293 T cells were transduced with the pCDH-CD39-GFP, psPAX2, and pMD2.G plasmids. The lentivirus was harvested and added to MC38 cells for 48 h. Transfected MC38 cells were sorted and seeded into 96-well plates, and plated as single clones. The overexpression of CD39 was confirmed by FACS staining.

### Bone marrow chimeric mice
C57BL/6 mice were lethally irradiated with Co60 at 10 Gy. The next day, $5 \times 10^6$ bone marrow cells from donor mice were intravenously transferred. WT/CD11c-DTR mixed bone marrow chimeras were generated by transplanting $1.5 \times 10^6$ WT and $3.5 \times 10^6$ CD11c-DTR bone marrow cells. P2X7R KO/CD11c-DTR mixed bone marrow chimeras were generated by transplanting $1.5 \times 10^6$ P2X7R KO and $3.5 \times 10^6$ CD11c-DTR bone marrow cells. The chimeras were administered prophylactic water containing antibiotics for four weeks following irradiation. 8–10 weeks after bone marrow transplantation, the mice were used for subsequent operations. For DC depletion, chimeric mice were intraperitoneally administered DT (Sigma) at a dose of 5 ng/g body weight 24 h before CV-1 treatment.

### Tumor growth and treatments
$5 \times 10^5$ Wild-type MC38, Atg5 KO, Ndufs6 KO, Isg15 KO, or MC38-CD39 cells were subcutaneously inoculated into the right flanks of

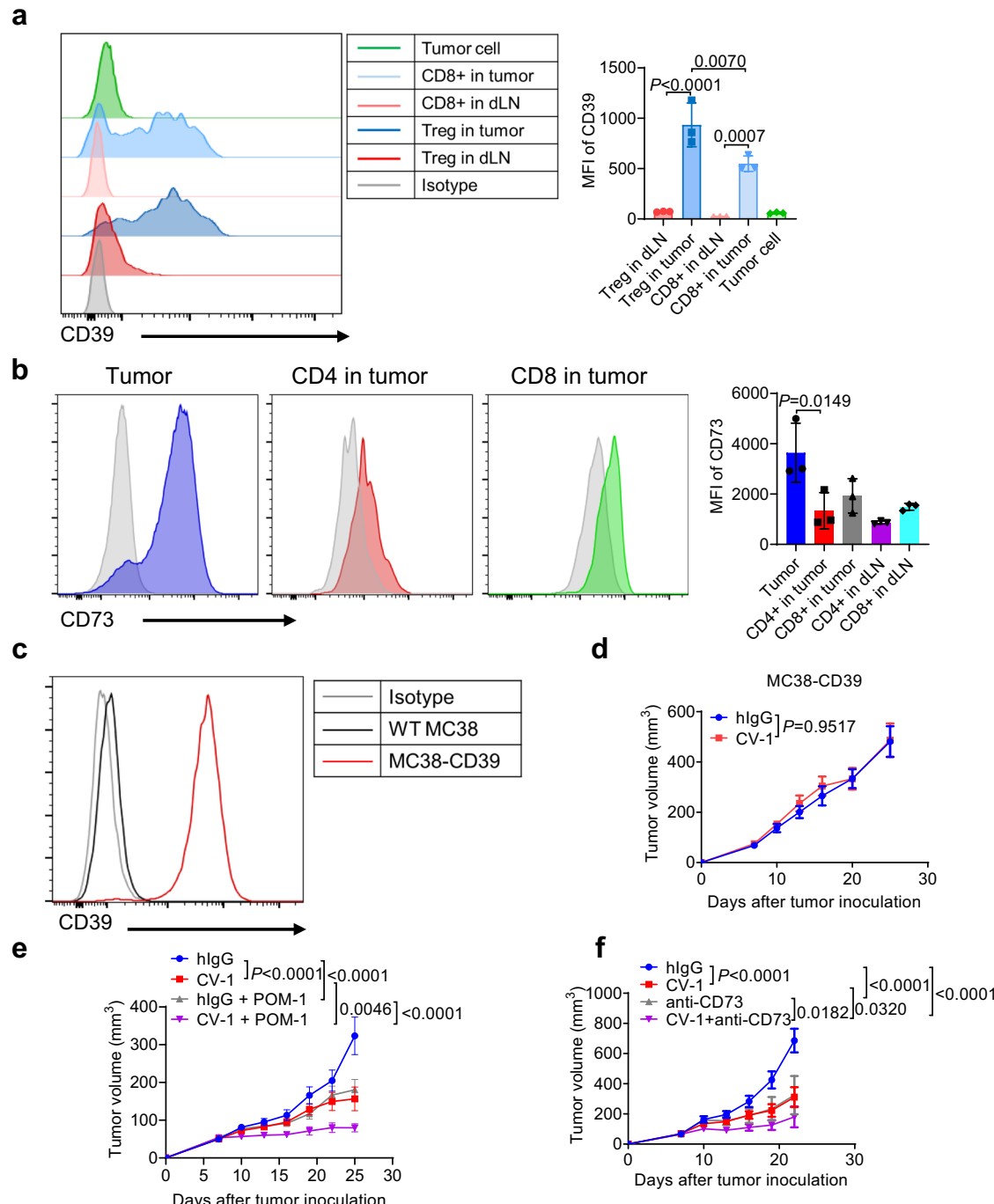

**Fig. 9 | CD47 blockade synergizes with CD39 or CD73 inhibition. a** The expression of CD39 on Treg cells, CD8 + T cells and tumor cells was measured ($n$ = 3 mice per group). **b** The expression of CD73 on CD4 + T cells, CD8 + T cells and tumor cells was measured ($n$ = 3 mice per group). **c** MC38 cells overexpressing CD39 was constructed and confirmed by FACS. **d** C57BL/6 mice ($n$ = 6 mice per group) bearing MC38-CD39 tumors were treated i.t. with CV-1 or hIgG every three days. Tumor volume was measured at indicated time. **e** C57BL/6 mice ($n$ = 6 mice per group) bearing MC38 tumors were treated i.t. with CV-1 or hIgG every three days.

POM-1 was administrated i.p. daily. Tumor volume was measured at indicated time. **f** C57BL/6 mice ($n$ = 6 mice per group) bearing MC38 tumors were treated i.t. with hIgG, CV-1, anti-CD73 mAb or CV-1 plus anti-CD73 mAb every three days. Data are representative of two independent experiments in (**a**)–(**d**), three independent experiments in (**e**) and (**f**). One-way ANOVA and multiple comparisons test was used in (**a**) and (**b**). Two-tailed unpaired Student's $t$ test was used in (**d**). Two-way ANOVA and multiple comparisons test were used in (**e**) and (**f**). Data are presented as mean values ± SEM. Source data are provided as a Source Data file.

C57BL/6 mice. Owing to the slower growth of tumors, 1 × 10⁶ IFNAR1 or IFNGR1 KO MC38 cells, 1×10⁶ CT26 cells, 3 × 10⁶ IFNAR1 KO CT26 cells, 1 × 10⁶ IFNGR1 KO CT26 cells, 1 × 10⁶ A20 cells, 3 × 10⁶ IFNAR1 KO A20 cells or 4 × 10⁶ IFNGR1 KO A20 cells were inoculated to achieve comparable tumor volumes before therapeutic treatment. Mice were treated intratumorally with 50 μg hIgG, CV-1, RatIg, or

Miap301 in combination with or without PPADS (50 μg), BzATP (1 pmol), IL-1Ra (200 μg), or anti-CD73 mAb (20 μg) every three days for four times. The total injection volume was 100 μL. POM-1 (300 μg) was administrated intraperitoneally daily. For CD8 or CD4 depletion experiments, 200 μg of anti-CD8 antibody (clone TIB210) or anti-CD4 (clone GK1.5) was administrated intraperitoneally twice a week.

Tumor growth was measured every three days and the volume was calculated as Length×Width×Height/2. The maximal tumor burden permitted by the ethics committee is no more than 1.5 cm in Length, Width and Height. In some cases, the limit has been exceeded the last day of measurement and the mice were immediately euthanized.

## ELISPOT

A total of 5 μg/ml purified anti-mouse IFN-γ was coated onto 96-well ELISPOT plates at 4 °C overnight (BD Biosciences). After blocking with a blocking buffer (RPMI 1640 medium contain 10% FBS), $5 \times 10^5$ CD45+ cells were sorted from tumors and stimulated with MC38 tumor cell lysis in 100 μl complete RPMI 1640 medium. After a 48 h incubation at 37 °C, tumor-specific IFN-γ producing cells were analyzed using a biotinylated anti-mouse IFN-γ and streptavidin-HRP (BD Biosciences). The information of antibodies used is listed in Supplementary Table 2. The spots were visualized with an AEC substrate set (BD Biosciences) and quantified with an auto-analyzing system.

## Flow cytometry and cell sorting

Apoptotic cells were stained with annexin V and 7-AAD for 15 min at 4 °C in annexin V binding buffer (BioLegend). Tumors and draining lymph nodes were isolated and digested into single-cell suspensions using collagenase I (1 mg/mL) and DNase (500 U/mL). CD8 + T cells were defined by CD45 + CD8+, CD4 + T cells were defined by CD45 + CD4+, Treg cells were defined by CD45 + CD4+Foxp3+, naïve OT-I T cells were defined by CD8 + TCRVa2 + CD62L + CD44−, DCs were defined by DAPI-MHCII+CD11c+ by flow cytometry. The information of antibodies used is listed in Supplementary Table 2. The samples were analyzed on an LSRFortessa flow cytometer (BD Biosciences). Cells were sorted on an Aria III flow cytometer (BD Biosciences). The data were analyzed using FlowJo v.10 software.

## ATP measurement

$5 \times 10^4$ tumor cells were seeded into 24-well plates and treated with or without 1 μg/mL recombinant IFN-α or 100/500 ng/ml recombinant IFN-γ. The medium and cell lysates taken from cells under specified conditions were immediately moved to ice and centrifuged at $500 \times g$ for 5 min. The supernatant was used for ATP detection using a luciferase-based kit (Beyotime) according to the manufacturer's protocol.

For in vivo ATP measurement, a mixture composed of luciferase-luciferin (Beyotime) was injected intratumorally in a volume of 40 μL. Luminescent images were obtained with a constant exposure time of two minutes on IVIS Lumina system (Perkin-Elmer). Regions of interest were manually defined around the tumor sites to determine the average radiance (p/s/cm²/sr).

## Seahorse assay

$4 \times 10^4$ MC38 cells were seeded into a cell culture plate and treated with or without 1 μg/mL recombinant IFN-α. Twenty-four hours later, samples were analyzed on a Seahorse XFe24 (Agilent) to measure OCR or ECAR. For OCR detection, oligomycin (1 μM), FCCP (1 μM), rotenone (1 μM), and Antimycin A (1 μM) were added. Glucose (10 mM), oligomycin (1 μM), and 2-DG (100 mM) were added for ECAR detection.

## mtDNA copy number measurement

$1 \times 10^5$ MC38 cells were seeded into 12-well plates and treated with or without 1 μg/mL recombinant IFN-α. Forty-eight hours later, the total DNA was extracted using the phenol-chloroform method. Quantitative real-time PCR was performed using genomic DNA primers (Hk2) and mtDNA primers (Nd1). The mtDNA copy number was calculated as $2 \times 2^{CT(Hk2)-CT(Nd1)}$.

## Mitochondrial mass, membrane potential, and ROS measurement

$1 \times 10^5$ MC38, A20, and HT29 cells were seeded in 12-well plates and treated with or without recombinant IFN-α. Forty-eight hours later, cells were stained with Mitotracker Green (20 nM, YEASEN), Mitotracker Deep Red (20 nM, YEASEN), and Mitotracker CMXRos (50 nM, Invitrogen) for 15 min at room temperature. Cells were washed with PBS containing 2% FBS and analyzed using an LSRFortessa flow cytometer (BD Biosciences).

## Immunofluorescence microscopy

For autophagosome detection, $1 \times 10^5$ MC38-GFP-LC3 cells were seeded in glass-bottom dishes and cultured overnight. Fresh medium with or without 1 μg/mL recombinant IFN-α was added. Autophagosome formation was analyzed using a confocal microscope (Zeiss LSM-710). The images were processed using ZEN 2010 software (Carl Zeiss, Inc.), and the GFP+ dots were analyzed using ImageJ 1.51j8 software.

For ATP-containing vesicle analysis, $1 \times 10^5$ wild-type or Atg5 KO MC38 cells were seeded into 24-well plates covered with glass sides and cultured overnight. Fresh medium with or without 1 μg/mL recombinant IFN-α was added. After 48 h, cells were cultured with 25 μM Quinacrine dihydrochloride (TRC) for 20 min at 37 °C in the dark. The cells were fixed with 4% paraformaldehyde and stained with DAPI for 10 min. Fluorescence images of quinacrine (green) and DAPI (blue) were obtained using a confocal microscope (Zeiss LSM-710). The images were processed using ZEN 2010 software (Carl Zeiss, Inc.), and the ATP-rich dots were analyzed using ImageJ 1.51j8 software.

## Western blot

For LC3 detection, MC38 cells cultured in six-well plates were treated with 1 μg/mL recombinant IFN-α or 100 ng/ml recombinant IFN-γ for 48 h with or without 50 μM chloroquine (CQ) for 24 h. For pSTAT1 and IRF1 detection, MC38 cells cultured in six-well plates were treated with 1 μg/mL recombinant IFN-α or 100 ng/ml recombinant IFN-γ for 48 h. $10^6$ cells were lysed with 80 μl lysis buffer (50 mM Tris HCl, pH 7.8, 150 mM NaCl, 2 mM EDTA, 1% Triton X-100, 0.1% SDS, Protease Inhibitor Cocktail (Beyotime)) on ice for 30 min. The lysate was then centrifuged at 12,000 rpm (9890 g) for 5 min at 4 °C and the precipitate was discarded. The supernatant was boiled for 10 min with loading buffer. For ISG15, ATG5 detection, $10^6$ cells were collected in 100 μl PBS and boiled for 10 min with loading buffer. 20-30 μl of the samples were loaded onto a 15-well, 12.5% SDS-PAGE and run at 100-120 V. The proteins were then transferred onto the activated PVDF at 260 mA for 90 min. The membrane was soaked in 5% w/v BSA to block non-specific binding sites, for 2 h at room temperature. Next, a solution containing specific primary antibody (prepared in 5% w/v BSA) at an appropriate dilution was added onto the membrane for overnight incubation at 4 °C. The membranes were then washed five times with TBST. The secondary antibody, also prepared in 5% w/v BSA, was incubated with the membrane for 1 h at room temperature. The membranes were then washed again with TBST five times and developed using Chemiluminescent HRP Substrate (Millopore). The information of antibodies used is listed in Supplementary Table 2. Images were captured by ChampChemi Gel Imaging System (SINSAGE). Densitometric quantitation of bands was performed using ImageJ 1.51j8 software.

## Quantitative real-time PCR

RNA from cultured MC38 cells and BMDM was extracted using an EasyPure RNA kit (TransGen), according to the manufacturer's protocol. RNA from sorted tumor cells and immune cells was extracted using the TRIzol reagent. The quality and quantity of total RNA were assessed using a Nanodrop spectrophotometer (ND 2000C; Thermo Fisher Scientific). Quantitative real-time PCR was performed using SYBR

Premix Ex TaqTM mix (Takara) and the reactions were run on a real-time PCR system (QuantStudio 7 Flex). Primers used for real-time PCR are listed in Supplementary Table 3.

## RNA-seq

C57BL/6 mice were inoculated subcutaneously with $5 \times 10^5$ MC38 cells. Seven days later, the tumor-bearing mice were treated intratumorally with 50 μg CV-1 or hIgG every three days. Two days after the third treatment, CD45− cells were sorted for RNA-seq. For the in vitro assay, $5 \times 10^5$ MC38 cells were seeded into six-well plates and treated with or without 1 μg/mL recombinant IFN-α. After 6, 24, and 48 h, the cells were harvested for RNA-seq. Samples were sequenced on an Illumina NovaSeq using a 2 × 150 bp paired-end run (BerryGenomics).

## Statistical analysis

Statistical significance was assessed using GraphPad Prism 9. Two-way ANOVA with multiple comparisons test, one-way ANOVA with multiple comparisons test, or two-tailed unpaired Student's $t$ test, was used as stated. RNAseq datasets were analyzed with DESeq2 package[89], with Negative Binomial GLM fitting and Wald statistics, two-sided, Benjamini–Hochberg was chosen for multiple comparisons adjustment. The results are expressed as mean ± SEM. A value of $P < 0.05$ was considered statistically significant.

## Reporting summary

Further information on research design is available in the Nature Portfolio Reporting Summary linked to this article.

## Data availability

The RNA-seq data generated in this study have been deposited in the NCBI GEO DataSets with GEO accession GSE235113. The remaining data are available within the Article, Supplementary Information or Source Data file. Source data are provided with this paper.

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

## Acknowledgements

We thank Prof. Jinhui Tao (The First Affiliated Hospital of University of Science and Technology of China) for generously providing P2X7R knockout mice, Prof. Taotao Wei (Institute of Biophysics, Chinese Academy of Sciences) for kind help on Seahorse assay. This work was supported by grants from the National Natural Science Foundation of China (82025015 and 32230038 to M.Z., 32000653 to W.W., and 81901688 to M.L.), Beijing Nova Program (Z201100006820032 and Z211100002121149 to W.W.), National Key R&D Program of China (2019YFA0905903 to M.Z. and 2022YFC2304104 to W.W.), and Strategic Priority Research Program of the Chinese Academy of Sciences (XDB29040202 to M.Z.).

## Author contributions

H.Z., W.W., and M.Z. designed the experiments and analyzed the data; H.Z. and W.W. conducted most experiments; H.X. and Y.L. prepared CV-1 and IFN-α recombinant proteins; J.D and B.R helped with RNA-seq analysis; M.L. helped with the Seahorse assay and ATP measurement; H.P. and Y.X.F. provided reagents; Y.X.F. provided advice on project design and reviewed the manuscript; H.Z., W.W., and M.Z. wrote the manuscript; and M.Z. conceived and supervised the project.

## Competing interests

The authors declare no competing interests.
