## [Peer Review File · Nature Communications]

Metabolic reprogramming mediated by tumor cell-intrinsic type I IFN signaling is required for CD47-SIRP α blockade efficacyREVIEWER COMMENTS

Reviewer #1 (Remarks to the Author): with expertise in cancer immunology, IFNs

The study by Zhou et al. investigated the role of tumor cell-intrinsic IFN signaling in innate CD47-SIRP α blockade immunotherapy. Using the transplant MC38 colorectal cancer model the authors demonstrate that tumor cell-intrinsic type I IFN signaling, in contrast to type II (IFN γ) signaling, is of importance in controlling tumor growth upon intratumoral application of SIRP α -hlg (CV-1). They provide data showing that type I IFN signaling switches the metabolism of MC38 towards oxidative phosphorylation, leading to enhanced ATP production and release of ATP-containing vesicles, with ISG15 and autophagy involved in the processes, respectively. Additional data indicate that ATP is needed for T cell priming and that combination of CV-1 with inhibitors of ATP-degrading ectoenzymes (CD39, CD73) improved anti-tumor effects.

Overall, this is a highly interesting and important study, providing novel mechanistic insights into innate CD47-SIRP α blockade immunotherapy. However, in the preclinical model some important data/controls, supporting the authors conclusions, are missing, Moreover, the clinical data included in the study are not very meaningful.

- To demonstrate the broader relevance of their findings, the authors should demonstrate the essential role of tumor cell-intrinsic type I IFN signaling (but not IFN γ signaling) in the context of CD47-SIRP α blockade immunotherapy in at least in one additional murine tumor model.
- The authors repeatedly state that CD8 T cells are important players in CD47-SIRP α blockade immunotherapy, referring to a paper published in Nature (PMID26322579; reference 9). However, in that previous study the importance of T cells was demonstrated in the A20 lymphoma model but not in the MC38 colorectal cancer model. Thus, data on the role of T cells in CD47-SIRP α blockade immunotherapy of transplanted MC38 tumors are lacking and should be provided (Fig. 1).
- Data confirming knockout of specific genes in MC38 cells is completely lacking (IFNAR,

IFNGR1, ISG15, ATG5,). Knockout should be confirmed at the protein level and in case signaling cascades are affected, this should also be addressed, e.g. lack of JAK/STAT pathway activation in IFNGR1 knockout cells.

- The manuscript includes a lot of in vitro data generated on MC38-WT and specific MC38 knockout cells. The data are of importance, but should, in some cases, be supplemented by data from corresponding tumors grown in mice, e.g. elevated expression of ISG15 in tumor cells in response to CV-1 treatment.

- Survival curves for mouse experiments should be shown.

- Figure 2c: Data on the effect of single PPADS (ATP receptor agonist) treatment on MC38 tumor growth is missing.

- Fig. 5f: Data show the correlation between expression of ATG4c and ATG7 and prognosis of colorectal cancer patients. Comments: 1) information about THPA data analyses is completely lacking, 2) what about ATG5-specific survival curves? 3) Do data show cancer cell-specific expression. If not, then the data are not very meaningful as expression is based on all cell types present in the tumor and should be deleted. Anyway, the link to therapy-induced autophagy is missing.

- Fig. 6: Data on autophagy in different settings are not convincing. If an increase in LC3-II under specific treatment cannot be detected this does not indicate a lack of autophagy, as LC3-II could be degraded in the lysosome. Therefore, experiments need to be carried out in the presence of bafilomycin or chloroquine that inhibit the autophagic flux.

- Fig. 7: What about expression of ISG15 in response to IFN-II?

- Fig. 9: There is no information about antibodies and marker combinations used to define T cell subpopulations by flow cytometry

•ISG15 has already been described as metabolic regulator in different studies, also in the context of cancer cells. Some of these studies should be listed (e.g. Nat Commun PMID: 32472071; Clin Transl Med PMID: 35842904)

Reviewer #2 (Remarks to the Author): with expertise in cancer, ATP

The study examined the role and signaling mechanisms of tumor cells responsiveness to type 1 interferons (IFN-Is) in antitumor immunity, by combining blockade of the CD46-SIRPalpha axis that couples tumor cells and dendritic cells. The in vitro and in vivo experiments support the notion that IFN-Is directly reprogram tumor cell metabolism by activating OXPHOS and enhancing ATP production and release via autophagy. Extracellular ATP, released upon CD47-SIRP α blockade, induce dendritic cell activation via the P2X7 receptor and thereby prime T cells to mediate the anti-tumor property of the CD47-SIRP α inhibitor. Such a finding may provide new strategies to improve the efficacy of tumor therapy. The study however suffers a number of deficiencies, with the major ones highlighted below. As commented in detail, some of these deficiencies such as data reproducibility and over-simplified interpretation of the data considerably diminish the interest of the finding of the study.

1. This whole study mainly depends on one tumor cell line (MC38). While this reviewer can accept it is a valid or useful cell model as claimed by the authors, the significance of the finding is limited as a result. It is important to verify the major observations using other colorectal tumor cell line (such as HT29) and also using other types of non-colorectal tumor cells.

2. One worrying matter is the reproducibility of many experimental results described in the manuscript. Many important results from experiments, assumably performed at different times but using exactly the same conditions, are hugely variable. This includes the growth/size of tumors from WT cells (>3 times difference from ~200 in Fig.1c, ~400 in Fig2c to >600 in Fig.3h and Fig.4e), intracellular ATP concentrations (~10 times difference from ~1 in Fig.3f, ~4 in Fig.3d to ~10 in Fig.4b), and the basal level extracellular ATP concentrations (60 times difference! from ~0.1 in Fig.4c, ~1.0 in Fig.2a, 1.5~2 in Fig.7b and Fig.5d, to ~6 in

Fig.3g), OCR level (~350 in Fig.4d and 600 in Fig.3c), and LC3-II/LC3-I ratio (Fig.5b vs Fig.7e, particularly the much lower as shown in the representative blots in Fig.6c where no mean data are provided). Such huge variations for different experiments, with much smaller errors in the data described in each panel raise serious concerns, as well as being extremely confusing, to this reviewer.

3. On related issues, in many experiments where the figure legend indicates $n = 3$ groups, the variations are unbelievably small or one in cases (e.g., Fig.3a-b-c-d-e-f; Fig.4b-c-d; Fig.5d; Fig.6a-b; Fig.6a-c-d). Does $n = 3$ refer to three independent experiments? If they are from replicates from one single experiments, the statistical analysis is completely meaningless!

4. The results presented in several figures appear to indicate that genetic deletion of particular signaling molecule in tumor cells altered tumor growth, probably resulting from cell proliferation, which the authors have not examined all or examined incompletely. The authors need to quantify such genetic effects clearly and take account into the interpretation or discuss their implications to their conclusion. This includes genetic knockout of IFNAR (Fig.1d), IFNGR (Fig.1e), ISG15 (Fig.4e). In addition, the control experiments using WT cells for examining the effect of ATG5-KO are missing (Fig.5e), and ATG5-KO seem to impair ATP production (Fig.5c) as well as released ATP (Fig.1d), but there is no comment on why?

5. In Fig.1e and h, where experiments were performed on IFNGR-KO MC39 tumor cells, tumor growth was normal in the early stage (the first 14 or 10 days) but reversed, i.e., became smaller in the very late stage, noticeably differing from the tumor from WT tumor cells (Fig.1c and f). please explain why?

6. The extracellular ATP concentrations in the solid tumor environment can reach hundreds of micromolar concentrations. However, the authors reported ATP in the nanomolar concentration range, even less than 0.1 nM (Fig.4c), which are usually present in culture medium anyway. This reviewer would like to question the resolution of the ATP assay used that enables them to detect ATP below 1 nM! Furthermore, do the authors really believe that ATP at such concentrations can activate the P2X7 receptor?

7. According to the conclusion, as illustrated in the graphic abstract, CV-1 induces inhibition of tumor growth via release of ATP from tumor cells and ATP in turn induces DC activation via P2X7 receptor; this notion is consistent with the results from experiments using the P2X7-KO (Fig.8d), where loss of P2X7 receptor led to no inhibition by CV1. One would expect that activation of the P2X7 receptor, the downstream signaling molecule, by BzATP, can circumvent the upstream signaling pathways and result in the same inhibitory effect on tumor growth as CV-1. However, in Fig.2d, it was shown that administration of BzATP failed to inhibit tumor growth in the absence of CV-1, why?

8. The study has not addressed the important role of macrophages residing in the tumors and interact with tumor cells, particularly activation of the P2X7 receptor expressed in macrophages in inhibiting tumor growth, reported by several previous studies examine many types of tumors/cancers including colorectal cancer. Fig.6b shows released ATP from BMDM, particularly considering that the P2X7-KO mice are lacking of the P2X7 receptor in macrophages as well as in DC. The authors should further explore the implications of this observation, which seems not fit the mechanism proposed by the authors. In Fig.6c, the basal autophagy level in BMDM, based on the LC3-II/LC3-I although it is fully justified, was pretty high, and treatment with IFN-alpha seems to be elevated the autophagy level if the ECL exposure was longer. Finally, the experiments shown in Fig.6c-d needs to be repeated to obtain the mean data with adequate statistical analysis.

9. The results from experiments showed Fig.9 are interesting. However, how mechanistically CV-1 inhibits tumor growth in synergy with CD39 and CD73, namely, how CV-1 alters the activities of these ATP-degrading enzymes require to be clarified?

10. Finally, the authors applied Student's t-test (or ANOVA) to compare the tumor growth, which is inappropriate. It is complete wrong to apply Student's t-test to compare the results of three groups or more (e.g., Fig.2a; Fig.3f; Fig.4b-c-d-e; Fig.5c-d; Fig.7a-b-c-d-e-f; Fig.8b-c; Fig.9a-b).

11. Finally, the authors used numerous misleading or confusing words, phrases or terms throughout the manuscript. For example, the phrase "blockade therapy" used in many

places including subheadings, which however has no bearing at all in this study that is restricted to mice. What does “immunogenic apoptosis” mean? “...products of OXPHOS, such as ATP and ROS, have been shown to promote anti-tumor responses”: ATP can promote tumor growth, via activation of the P2X7 receptor in tumor cells and at the same time inhibit tumor growth by activation of the P2X7 receptor in tumor-residing immune cells! In addition, the authors failed to provide the reference or evidence to support the statements in many parts, e.g., “Extracellular ATP release is actively regulated via autophagy upon chemical drug treatment”, “IFN-Is induce autophagy and ATP release in tumor cells”, “maintaining a high ATP concentration by preventing its degradation into immunosuppressive adenosine may facilitate anti-tumor immunity induced by CD47 blockade”, “We also found that CD47-SIRP α ICB efficacy depended on IL-1 β ”?

Reviewer #3 (Remarks to the Author): with expertise in cancer immuno-metabolism

Comments to:

Tumor cell intrinsic type I interferon signaling mediated metabolic remodeling dictates CD47-SIRP α blockade immunotherapy
Zhou et al. for Nature Communications

Summary

In this manuscript, Zhou et al elaborates on the mechanistic understanding of CD47-SIRP α immune blockade therapy. Using RNA-seq on tumor cells exposed to SIRP α -hIg (CV-1) they find that IFN-I signaling, and autophagy is upregulated after CV-1. They delineate a pathway where IFN-I activates IFNAR and Isg15 on tumor cells, resulting in increased OXPHOS and autophagy further leading to release of ATP. Using CRISPR/Cas9 guided KO of tumor cells, they show that tumor cells without Isg15, Ndufs6 (OXPHOS) or Atg5 (Autophagy) are unresponsive to CV-1 therapy.

Zhou et al argue that CV-1 promotes T cell priming through this mechanism, since injecting CV-1 in IFNAR $^{-/-}$ tumors did not have an effect on IFNAR $^{-/-}$ or a distant WT tumor in the same mouse, whereas injecting CV-1 into WT tumor, suppressed distant IFNAR $^{-/-}$ tumors in the same mouse, and CV-1 increased CD8 $^{+}$ T cell percentage in an IFNAR-dependent way. They use mixed bone marrow chimera of ATP receptor P2X7R $^{-/-}$ and CD11c-DTR mice to

show that mice without P2X7R in DCs become resistant to CV-1 treatment, and blocking IL-1b with IL-1Ra abolished CV-1 effect.

Lastly, they show that overexpression of ATP degrading enzyme CD39 by tumor cells, abolishes the effect of CV-1 on tumor growth, whereas CD39 inhibitor or CD73 antibody enhanced anti-tumor efficacy in combination with CV-1, supporting that high extracellular ATP facilitate anti-tumor response by CD47 blockade.

This work presents novel tumor cell intrinsic mechanistic findings that expands our understanding of CD47-SIRPa blockade therapy using thorough experimental approaches. A few limitations can be found in the immunological investigations which are listed below for further revision.

Major comments

1. In this manuscript they show that IFN-I induces OXPHOS and autophagy, which are both required for ATP release. Yet, how these two are associated is not clear. Could the authors show if mitophagy is also regulated in this setting, and if that is relevant for IFN-I-induced ATP release?
2. The authors claim that tumor cell intrinsic IFN-I signaling is required for T cell priming after CV-1, yet this is only supported by increased CD8+ T cell percentages in the tumor. To confirm that this is due to priming other experiments are needed. For example by ex vivo co-culture assays with CV1/ATP-treated APCs.
3. CD8 T cell percentages are increased, but the functionality of the T cells is not investigated. Killing assay or cytokine release assays of CD8 T cells is necessary to state that they are increased in functionality.

Note: In the result section it is written "IFNAR1 deficiency in tumor cells does not influence the effector T cell killing function", yet without any figure reference or reference to literature.

Minor comments

1. IFN-I increases both mitotracker green (MDR) and mitotracker deep red (MDR) suggesting an increased mitochondrial biosynthesis. To give an expression of the qualitative differences

of mitochondria, the authors should also show mitochondrial functionality normalized to mitochondria mass (MDR/MG).

2. In figure 4e it looks like ISG15 KO MC38 grows slower than WT, possibly explaining why the responsiveness is ablated in these tumors. Can the authors comment in this?

3. CD8 T cell percentages are increased in tumors after CV-1 injection, however the count of CD8 T cells normalized to tumor weight is also relevant to show.

Reviewer #4 (Remarks to the Author): with expertise in cancer, CD47

Tumor metabolic rewiring plays a critical role in cancer immunotherapy, however, some details in the metabolic features in the tumor microenvironments remain uneducated. In this study, Zhou et al investigated IFN-Is, pleiotropic cytokines enriched in the tumor microenvironment, in tumor metabolic rewiring with OXPHOS-mediated resistance to anti-CD47 therapy. The results indicate that tumor responsiveness to IFN-Is is essential for CD47-SIRP α blockade immunotherapy; IFN-Is can guide the tumor metabolic rewiring with OXPHOS activation that enhances cellular ATP generation which together with autophagy-derived ATP can raise the extracellular ATP pool leading to enhanced tumor cell response to anti-CD47 therapy. ATP released upon CD47-SIRP α blockade primes the anti-tumor T cell response via ATP-P2X7 receptor-mediated dendritic cell activation which is supported by ATP-degrading ectoenzymes, CD39 and CD73. These results reveal a new mechanism of enhanced OXPHOS that can boost tumor response to CD47-targeted ICT.

The overall experimental design is reasonable, and the major data are supportive of the conclusion which could contribute to a new version of metabolism-associated tumor response to ICT, thus holding a potential clinic benefit. However, in addition to scientific deficiency including the lack of macrophagic analysis, there are some major concerns about professional data presentation and explanation. The overall scientific writing is to be substantially improved. A focus and deepening the analysis of the diet using the in vivo test would be appreciated and informative.

Scientific Comments:

1. The specific pathways of IFN-Is activate tumor OXPHOS which are believed to be deficient

in many solid tumor cells (Warburg Effect). Does shifting from glycolysis to OXPHOS dominant pathway occur in the tumor model used in this work?

2. It is not clear whether tumor resistance to CD47-SIRPα blockade is due to the reduced or enhanced copy numbers of CD47 receptors. Does the CD47 gene copy number enhanced or reduced by mitochondrial OXPHOS?

3. In addition to ATP generation and release, OXPHOS can produce many other biological effects which can alter cell survival rate with or without ICT, which should be discussed.

4. In Fig. 1, it is unclear why need to sort the tumor cells after in vivo tumor treatment with CD47-SIRPα blockade, and what is the control transcriptomics compared in the treated tumor cells.

Scientific Presentation

5. The overall writing, especially on the precise description of experimental procedures is in need; especially a critical review by senior authors is mandatory (Dr. YX Fu, a well-known expert in the field is listed as one of the authors but not involved in the paper writing).

6. In the introduction, critical reading is required for improving the scientific language and paper citation, such as “In recent years, the innate immune checkpoint has emerged as a popular target for cancer immunotherapy in recent years. The CD47-SIRPα axis is the best-studied innate checkpoint in cancer. Therapies designed to disrupt the CD47-SIRPα interaction are by far the furthest along the clinical development process and are now being investigated for multiple human cancers”. This description does not reflect the current anti-CD47 and is not precisely related to the PD-1/PD-L1 blockage therapy.

7. As cited in ref 48, Isg15 has been reported to govern mitochondrial function in macrophages following vaccinia virus infection (ref 48) which found that “Post-translational modifications such as ISGylation regulate essential mitochondrial processes including respiration and mitophagy, and influence macrophage innate immunity signaling”. A further digger into the Isg15-mediated specific elements in the mitochondria respiration such as the complexes phosphorylation would be informative.

8. The first sentence of the Results “To investigate tumor cell responsiveness during immunotherapy (what is the specifically immunotherapy indicated here?), established MC38 tumors (what are the tumors used here, should be indicated here) were treated with SIRPα-hlg (CV-1) (what is this, antibody?), a high-affinity mutant for CD47 binding”.

Dear Reviewers,

We really appreciate the valuable and insightful comments/suggestions on our manuscript. We are encouraged with the overall positive comments. To address all remaining concerns, we have conducted a lot of new experiments, reanalyzed the data, reorganized the figures, and extensively edited throughout the text. The major revisions have been particularly highlighted in the text.

Please see below for the point-by-point responses in more details. We hope that our revised manuscript is now suitable for publication.

Reviewer #1 (Remarks to the Author): with expertise in cancer immunology, IFNs

The study by Zhou et al. investigated the role of tumor cell-intrinsic IFN signaling in innate CD47-SIRP α blockade immunotherapy. Using the transplant MC38 colorectal cancer model the authors demonstrate that tumor cell-intrinsic type I IFN signaling, in contrast to type II (IFN γ) signaling, is of importance in controlling tumor growth upon intratumoral application of SIRP α -hIg (CV-1). They provide data showing that type I IFN signaling switches the metabolism of MC38 towards oxidative phosphorylation, leading to enhanced ATP production and release of ATP-containing vesicles, with ISG15 and autophagy involved in the processes, respectively. Additional data indicate that ATP is needed for T cell priming and that combination of CV-1 with inhibitors of ATP-degrading ectoenzymes (CD39, CD73) improved anti-tumor effects.

Overall, this is a highly interesting and important study, providing novel mechanistic insights into innate CD47-SIRP α blockade immunotherapy. However, in the preclinical model some important data/controls, supporting the authors conclusions, are missing. Moreover, the clinical data included in the study are not very meaningful.

- 1. To demonstrate the broader relevance of their findings, the authors should demonstrate the essential role of tumor cell-intrinsic type I IFN signaling (but not IFN γ signaling) in the context of CD47-SIRP α blockade immunotherapy in at least in one additional murine tumor model.**

Response:

This reviewer commented: “this is a highly interesting and important study, providing novel mechanistic insights...”. We highly appreciate this comment. Recently, the clinical investigation of CD47-SIRP α blockade therapy does not go smoothly, emphasizing the importance of further understanding the mechanisms of this blockade

therapy. Indeed, as the reviewer suggested, a broader relevance would be more convincing and insightful. We have therefore tested this mechanism in two other murine tumor models, CT26 and A20. In both models, tumor cell-intrinsic type I IFN signaling, but not type II signaling, is required for CD47-SIRP α blockade therapy. The data have been added as Supplementary Fig. 1g-l, and Supplementary Fig. 1m-r, respectively, also shown below.

2. The authors repeatedly state that CD8 T cells are important players in CD47-SIRP α blockade immunotherapy, referring to a paper published in Nature (PMID26322579; reference 9). However, in that previous study the importance of T cells was demonstrated in the A20 lymphoma model but not in the MC38 colorectal cancer model. Thus, data on the role of T cells in CD47-SIRP α blockade immunotherapy of transplanted MC38 tumors are lacking and should be provided (Fig. 1).

Response:

Thank you for this point. To confirm the importance of CD8⁺ T cells in CD47-SIRP α blockade therapy, we have performed the experiments in TCR deficient mice or in WT mice treated with anti-CD8 (TIB210) or anti-CD4(GK1.5) depleting antibodies. The results now clearly showed that T cells, especially CD8⁺ T cells is critically required for the therapy (Fig. 2a,b).

3. Data confirming knockout of specific genes in MC38 cells is completely lacking (IFNAR, IFNGR1, ISG15, ATG5, ...). Knockout should be confirmed at the protein level and in case signaling cascades are affected, this should also be addressed, e.g. lack of JAK/STAT pathway activation in IFNGR1 knockout cells.

Response:

- The knockout efficiency of IFNAR1 and IFNGR1 KO MC38 cell lines were confirmed by genomic sequencing at gene level (Supplementary Fig. 1a), FACS staining at protein level (Supplementary Fig. 1b) and JAK/STAT activation at functional level (Supplementary Fig. 1c).

- ISG15 KO MC38 cell line knockout efficiency was confirmed by genomic sequencing at gene level and Western Blot at protein level (Supplementary Fig.4b).

- ATG5 KO MC38 cell line knockout efficiency was confirmed by genomic sequencing at gene level and Western Blot at protein level (Supplementary Fig.5e).

4. The manuscript includes a lot of in vitro data generated on MC38-WT and specific MC38 knockout cells. The data are of importance, but should, in some cases, be supplemented by data from corresponding tumors grown in mice, e.g. elevated expression of ISG15 in tumor cells in response to CV-1 treatment.

Response:

Thanks for this important suggestion. We have performed qRT-PCR to detect the expression of *Isg15*, OXPHOS genes and autophagy genes in MC38 tumor cells grown in mice with CV-1 treatment or control treatment. We found that the expression of these genes was all up-regulated upon CV-1 treatment. (Fig. 4b, Supplementary Fig. 3k, Supplementary Fig. 5c)

5. Survival curves for mouse experiments should be shown.

Response:

The survival curves for the typical experiments have been shown, as in Fig. 1f-h, Supplementary Fig. 1g-i, and Supplementary Fig. 1p-r.

6. Figure 2c: Data on the effect of single PPADS (ATP receptor agonist) treatment on MC38 tumor growth is missing.

Response:

We are sorry for missing this control. The experiment has been repeated with this control included (Fig. 2g).

7. Fig. 5f: Data show the correlation between expression of ATG4c and ATG7 and prognosis of colorectal cancer patients. Comments: 1) information about THPA data analyses is completely lacking, 2) what about ATG5-specific survival curves? 3) Do data show cancer cell-specific expression. If not, then the data are not very meaningful as expression is based on all cell types present in the tumor and should be deleted. Anyway, the link to therapy-induced autophagy is missing.

Response:

We agree the clinical data shown may have alternative explanations. In addition, these data have no relationship with CD47-SIRPa blockade therapy. We actually have also tried to find some clinical data of CD47-SIRPa blockade in solid tumors, but failed since this kind of investigation is very rare. Therefore, we would like to follow this reviewer's suggestion to delete these data. Hopefully, our data would provide some guidance for future clinical investigation.

8. Fig. 6: Data on autophagy in different settings are not convincing. If an increase in LC3-II under specific treatment cannot be detected this does not indicate a lack of autophagy, as LC3-II could be degraded in the lysosome. Therefore, experiments need to be carried out in the presence of bafilomycin or chloroquine that inhibit the autophagic flux.

Response:

Thank you for this suggestion. To avoid this caveat, we have repeated the related experiments with or without chloroquine. The data showed similar pattern under two conditions, thus, the conclusions remain the same (Fig. 5b, Fig. 6c, and Fig. 7f).

9. Fig. 7: What about expression of ISG15 in response to IFN-II?

Response:

Isg15 also showed upregulation upon IFN-II, but only minor change (5.29-fold with

100ng/ml IFN γ and 5.73-fold with 500ng/ml IFN γ) compared to IFN-I stimulation (144.33-fold). This probably explain why IFN-I, but not IFN-II, could induce tumor cell OXPHOS and ATP production (Fig. 7c).

10. Fig. 9: There is no information about antibodies and marker combinations used to define T cell subpopulations by flow cytometry

Response:

We are sorry for missing relevant information.

- CD8 T cell was defined by CD45+ CD8+,
- CD4 T cell was defined by CD45+ CD4+,
- Treg cell was defined by CD45+ CD4+Foxp3+.

Antibody information see below.

- CD45 PerCP-5.5(Ly-5) (Clone: 30-F11; Cat. 45-0451; eBioscience)
- CD4 PE/Cy7 (Clone:GK1.5; Cat. 25-0041-82; eBioscience)
- CD8 APC-eFlourR 780 (Clone: 53-6.7; Cat. 47-0081-82; eBioscience)
- Foxp3 APC (Clone: FJK-16s; Cat. 17-5773-82; eBioscience)

We have added this information into the text and Reporting Summary.

11. ISG15 has already been described as metabolic regulator in different studies, also in the context of cancer cells. Some of these studies should be listed (e.g. Nat Commun PMID: 32472071; Clin Transl Med PMID: 35842904)

Response:

Thank you for providing these references. We have cited these studies to further support our conclusion.

Reviewer #2 (Remarks to the Author): with expertise in cancer, ATP

The study examined the role and signaling mechanisms of tumor cells responsiveness to type I interferons (IFN-I) in antitumor immunity, by combining blockade of the CD47-SIRP α axis that couples tumor cells and dendritic cells. The in vitro and in vivo experiments support the notion that IFN-I directly reprogram tumor cell metabolism by activating OXPHOS and enhancing ATP production and release via autophagy. Extracellular ATP, released upon CD47-SIRP α blockade, induce dendritic cell activation via the P2X7 receptor and thereby prime T cells to mediate the anti-tumor property of the CD47-SIRP α inhibitor. Such a finding may provide new strategies to improve the efficacy of tumor therapy. The study however suffers a number of deficiencies, with the major ones highlighted below. As commented in detail, some of these deficiencies such as data reproductivity and over-simplified interpretation of the data considerably diminish the interest of the finding of the study.

- 1. This whole study mainly depends on one tumor cell line (MC38). While this reviewer can accept it is a valid or useful cell model as claimed by the authors, the significance of the finding is limited as a result. It is important to verify the major observations using other colorectal tumor cell line (such as HT29) and also using other types of non-colorectal tumor cells.**

Response:

Thank you for the important advice. We have confirmed the dependency of type I IFN signaling, but not type II IFN signaling, in CD47-SIRP α blockade therapy in the other two in vivo murine mouse models, CT26 and A20 (Supplementary Fig. 1g-l, and Supplementary Fig. 1m-r). We have also confirmed some key findings using human tumor cell lines in vitro: in HT29 cells, IFN-I stimulation induces ATP production (Supplementary Fig. 3g), release (Supplementary Fig. 2e), mitochondria activity (Supplementary Fig. 3c-d), autophagy (Supplementary Fig. 5a), and ATP-rich vesical formation (Supplementary Fig. 5d). These data together suggest a general mechanism of the current finding.

- 2. One worrying matter is the reproducibility of many experimental results described in the manuscript. Many important results from experiments, assumably performed at different times but using exactly the same conditions, are hugely variable. This includes the growth/size of tumors from WT cells (>3 times difference from ~200 in Fig.1c, ~400 in Fig.2c to >600 in Fig.3h and Fig.4e), intracellular ATP concentrations (~10 times difference from ~1 in Fig.3f, ~4 in Fig.3d to ~10 in Fig.4b), and the basal level extracellular ATP concentrations (60 times difference! from ~0.1 in Fig.4c, ~1.0 in Fig.2a, 1.5~2 in Fig.7b and Fig.5d, to ~6 in Fig.3g), OCR level (~350 in Fig.4d and 600 in Fig.3c), and LC3-II/LC3-I ratio (Fig.5b vs Fig.7e, particularly the much lower as shown in the representative blots in Fig.6c where no mean data are**

provided). Such huge variations for different experiments, with much smaller errors in the data described in each panel raise serious concerns, as well as being extremely confusing, to this reviewer.

Response:

We are sorry for the misleading information. This project lasts for several years with several researchers involved. Therefore, the variation between experiments may be sometimes big although the exact condition was followed. The state of cell lines and mice, reagents used, personal standard for cell counting and tumor measurement, etc., may all have some influence on the actual value number of the results. However, the sample variation in the same experiment is usually small, which confirms the technical consistency of each individual performers. We hope this could be understood.

As to the specific concerns, pls see the explanation below.

- Tumor size:

Fig.1c and Fig2c (original manuscript) were performed at similar condition that the tumor size of hIgG group grows to about 200mm³ at day 25. Fig.2g (new version replaces Fig.2c in the original), Fig.3h and Fig.4f (Fig.4e in original manuscript) were performed recently, the tumor size of hIgG group grows to about 600mm³ at day 25. Comparing the growth curve of Fig2c (original manuscript) and Fig.2g (new version replaces Fig.2c in the original), tumor growth rate may indeed increase in recent experiments but this does not change the conclusion.

- Intracellular ATP concentrations:

Fig.3f shows the concentration of intracellular ATP per 10⁵ cells, but Fig.3d and Fig.4b show every 10⁶ cells, which result in ~10 times difference. We are sorry for this confusion and have unified the units.

- Extracellular ATP concentrations:

The ATP is not stable in vitro and the detection assay is very sensitive. At the nanomolar level of extracellular ATP concentrations, the manipulation of cell centrifugation, dissociating cells by pipette, total processing time, etc. may all lead to some differences among experiments. However, the sample variation in each individual experiment is usually kept small, confirming the technical consistency of each individual performers.

- OCR level:

Fig.4d and Fig.3c were performed by different researchers at different times (interval more than one year). In addition, the difference between 350 in Fig.4d and 600 in Fig.3c are not considered huge.

- LC3-II/LC3-I ratio:

Fig.5b, Fig.6c and Fig.7f have all been repeated recently and updated.

- 3. On related issues, in many experiments where the figure legend indicates $n = 3$ groups, the variations are unbelievably small or one in cases (e.g., Fig.3a-b-c-d-e-f; Fig.4b-c-d; Fig.5d; Fig.6a-b; Fig.6a-c-d). Does $n = 3$ refer to three independent experiments? If they are from replicates from one single experiments, the statistical analysis is completely meaningless!**

Response:

$n = 3$ refers to three biological replicates from one representative experiment. Each experiment was repeated at least twice. Data are representative of at least two independent experiments. We followed the requirement of Unpaired Student's t-test to do the statistical analysis. Statistical analysis is very important. This issue is also addressed in more details in our response to point #10 of this reviewer.

- 4. The results presented in several figures appear to indicate that genetic deletion of particular signaling molecule in tumor cells altered tumor growth, probably resulting from cell proliferation, which the authors have not examined all or examined incompletely. The authors need to quantify such genetic effects clearly and take account into the interpretation or discuss their implications to their conclusion. This includes genetic knockout of IFNAR (Fig.1d), IFNGR (Fig.1e), ISG15 (Fig.4e). In addition, the control experiments using WT cells for examining the effect of ATG5-KO are missing (Fig.5e), and ATG5-KO seem to impair ATP production (Fig.5c) as well as released ATP (Fig.1d), but there is no comment on why?**

Response:

- The genetic deletion of *Ifnar1*, *Ifngr*, *Atg5*, *Ndufs6* and *Isg15* in tumor cells indeed altered tumor growth rate in vivo to certain degrees. Owing to the slower growth of these KO tumor cells, twice as many or even more cells were used to inoculate the mice to acquire comparable tumor volumes before therapy, see the Methods section. It is assumed that slower tumor growth may be easier for treatment. However, *Ifnar1*, *Atg5*, *Ndufs6* and *Isg15* deficiency in MC38 tumor cells almost completely resist the therapeutic effect of CV-1, which further confirmed our conclusions.
- The control experiment using WT MC38 has been added in Fig.5e. We are sorry for missing this.

- Yes, our data suggest that Atg5 deficiency may be also required for IFN-I increased intracellular ATP production to some degree (Supplementary Fig. 5f). The exact reason is still unclear. A previous study has reported that Atg5-deficient dendritic cells showed increased activation of glycolysis and decreased mitochondrial activity (Autophagy. 2021 Sep;17(9):2111-2127.). Whether Atg5 deficiency may alter the metabolic state of MC38 cells remains to be confirmed in future. Thank the reviewer for pointing this out. We have also commented this in the text.

5. In Fig.1e and h, where experiments were performed on IFNGR-KO MC38 tumor cells, tumor growth was normal in the early stage (the first 14 or 10 days) but reversed, i.e., became smaller in the very late stage, noticeably differing from the tumor from WT tumor cells (Fig.1c and f). please explain why?

Response:

This is probably a visual effect. Please note that the y-axis for tumor size is very different between these two tumors. WT MC38 tumors grew much faster than IFNGR-KO MC38 tumors. Upon CV1 or Miap301 treatment, both types of tumors responded largely similarly at the beginning. However, the key difference is, at later time, that WT MC38 tumors grew slowly but still continuously increased in size, while IFNGR-KO MC38 tumors even shrunk. This difference may cause a visual effect as the reviewer mentioned. This also suggests that IFNGR-KO MC38 tumors may be more easily controlled by CD47 blockade than WT MC38 tumors.

6. The extracellular ATP concentrations in the solid tumor environment can reach hundreds of micromolar concentrations. However, the authors reported ATP in the nanomolar concentration range, even less than 0.1 nM (Fig.4c), which are usually present in culture medium anyway. This reviewer would

like to question the resolution of the ATP assay used that enables them to detect ATP below 1 nM! Furthermore, do the authors really believe that ATP at such concentrations can activate the P2X7 receptor.

Response:

Thank the reviewer for raising this important issue. As to the extracellular ATP assay, an Enhanced ATP Assay kit (Beyotime, S0027) was used in our study. The linear detection range of the kit is 0.1nM-10mM. Thus, we think the concentration we detected should be valid. However, we do agree that such minor concentration may not be able to activate P2X7 receptor efficiently. The extracellular ATP concentrations measured in the cell culture medium maybe underestimated since ATP is not stable under room temperature in vitro during the processing and measurement. Thus, while the absolute concentration may not be accurate, the comparison is still valid. Our current study suggests that IFN-I treatment could induce higher extracellular ATP release compared to the untreated group. In addition, the in vivo concentration could be very different. Although we have shown that in vivo CV-1 treatment induced extracellular ATP in tumor tissues at a higher level than that in control treatment (Fig. 2f), a more accurate quantitative in vivo detection assay is still required in future, which is a limitation of our current study.

- 7. According to the conclusion, as illustrated in the graphic abstract, CV-1 induces inhibition of tumor growth via release of ATP from tumor cells and ATP in turn induces DC activation via P2X7 receptor; this notion is consistent with the results from experiments using the P2X7-KO (Fig.8d), where loss of P2X7 receptor led to no inhibition by CV1. One would expect that activation of the P2X7 receptor, the downstream signaling molecule, by BzATP, can circumvent the upstream signaling pathways and result in the same inhibitory effect on tumor growth as CV-1. However, in Fig.2d, it was shown that administration of BzATP failed to inhibit tumor growth in the absence of CV-1, why?**

Response:

This indicates that CV-1 does more than ATP. In fact, in addition to ATP production, tumor cell phagocytosis is also promoted upon CV-1 treatment, which leads to enhanced tumor antigen processing and presentation. Without significant tumor antigen presentation (signal 1 for T cell response), ATP by itself is not enough. This may explain why BzATP alone has no inhibitory effect on tumor growth.

- 8. The study has not addressed the important role of macrophages residing in the tumors and interact with tumor cells, particularly activation of the P2X7 receptor expressed in macrophages in inhibiting tumor growth, reported by several previous studies examine many types of tumors/cancers including**

colorectal cancer. Fig.6b shows released ATP from BMDM, particularly considering that the P2X7-KO mice are lacking of the P2X7 receptor in macrophages as well as in DC. The authors should further explore the implications of this observation, which seems not fit the mechanism proposed by the authors. In Fig.6c, the basal autophagy level in BMDM, based on the LC3-II/LC3-I although it is fully justified, was pretty high, and treatment with IFN-alpha seems to be elevated the autophagy level if the ECL exposure was longer. Finally, the experiments shown in Fig.6c-d needs to be repeated to obtain the mean data with adequate statistical analysis.

Response:

Thank for raising this important point of who respond to eATP. The role of extracellular ATP/P2X7 receptor in tumor macrophages has been reported in several studies, however the conclusion is still controversial. Li et al. reported that anti-CD39 treatment induced tumor growth inhibition is dependent on host P2X7 receptor using P2X7-deficient mice [1]. This study hypothesized that anti-CD39 treatment triggers an eATP-P2X7-inflammasome-IL18 axis in tumor myeloid cells, probably macrophages, leading to enhanced intratumor T-cell effector function. However, since no conditional knockout mice were used, these results do not exclude other cells than myeloid cells. In a later study, Qin et al. using BMDM transfer murine model showed that P2X7 on macrophages promotes lung cancer growth likely via M2-polarization and reduced CD8+ T cell recruitment and activation [2]. This study suggests that eATP-P2X7 may even dampen anti-tumor response. Recently, Casey reported that pharmacological blockade of CD39 prevented eATP degradation and enhanced engulfment of antibody-coated lymphoma cells by macrophages in vitro in a P2X7 receptor-dependent manner, indicating that eATP fueled antibody-dependent cellular phagocytosis (ADCP) activity [3]. However, whether eATP-P2X7 activated ADCP is required for in vivo efficacy still remains to be determined. Thus, the role of eATP/P2X7 on tumor macrophages remained to be clarified.

It should be noted that ATP might also work on other cells inside tumors, including dendritic cells, T cells, and even tumor cells. In a previous study [4], P2X7 deficiency on hematopoietic cells was found to accelerate B16 melanoma growth, associated with reduced intratumor IL-1b and VEGF. Interestingly, dendritic cells from P2X7-deficient mice were unresponsive to stimulation with tumor cells for IL-1b production. This is consistent with our current finding of important role of dendritic cell P2X7 for CV-1 treatment. In our study, we have used conditional mouse model, but not global P2X7-deficient mouse model as used in previous studies, to selectively deplete P2X7 from dendritic cells and found it is required for CV-1 blockade efficacy (Fig. 8e). Thus, although our current study could not exclude the role of eATP/P2X7 on macrophages, T cells, or tumor cells for the efficacy of CV-1 treatment, it is clear that dendritic cell responsiveness to eATP via P2X7 is required for the efficacy.

The above discussion has been simplified and added into the Discussion section.

To specifically confirm the effect of IFN-I on myeloid cell autophagy, Fig.6c-d have been repeated and statistical analysis were added.

9. The results from experiments showed Fig.9 are interesting. However, how mechanistically CV-1 inhibits tumor growth in synergy with CD39 and CD73, namely, how CV-1 alters the activities of these ATP-degrading enzymes require to be clarified?

Response:

Yes, this is an interesting finding. Given current poor clinical performance of CD47-SIRP α blockade therapy, especially for solid tumors, this finding may provide a new direction for clinical study design. As the reviewer suggested, we examined whether CV-1 treatment would affect the expression of CD39 or CD73 in tumors. No change was found in the major cell types with CD39 or CD73 expression (Supplementary Fig. 7a,b). Although CD39 or CD73 in tumors was not changed by CV-1 treatment, it does not mean they are not working, given their abundant expression (Fig. 9a,b). CV-1 treatment-induced ATP could still be degraded to adenosine, which is suppressive. Thus, CV-1 treatment combination with CD39 or CD73 inhibitor would rescue more ATP for dendritic cell activation and T cell priming.

10. Finally, the authors applied Student's t-test (or ANOVA) to compare the tumor growth, which is inappropriate. It is complete wrong to apply Student's t-test to compare the results of three groups or more (e.g., Fig.2a; Fig.3f; Fig.4b-c-d-e; Fig.5c-d; Fig.7a-b-c-d-e-f; Fig.8b-c; Fig.9a-b).

Response:

We highly appreciate this comment. Biological statistics is a very very important step during the research. However, the fact is that it is not always properly used. It is very common that data from the same type of experiments were analyzed in various different ways in different research papers, even from prominent journals.

For instance, tumor growth comparison has been analyzed using following methods:

- Student's t-test for either two groups or more
<https://www.nature.com/articles/s41591-018-0023-9#Sec2>
- two-way ANOVA for either two groups or more
<https://www.nature.com/articles/s41586-023-06026-3>
<https://www.nature.com/articles/s43018-023-00600-4>
- Student's t-test for two groups and one-way ANOVA for three or more groups
<https://www.nature.com/articles/s41590-023-01723-7>

As to the bar graphs which was extensively used in current study, similar diversity exists:

- Student's t-test for either two groups or more
<https://www.nature.com/articles/s41591-018-0023-9#Sec2>
<https://www.nature.com/articles/s41586-021-04057-2>
- Student's t-test for two groups and one-way ANOVA for three or more groups
<https://www.nature.com/articles/s43018-023-00600-4>
- Mann-Whitney test for two groups and one-way or two-way ANOVA for three or more
<https://www.nature.com/articles/s41586-023-06026-3>

Obviously, choosing a proper statistical test sometimes is not simple. We have carefully analyzed these samples papers and also sought professional statistical guideline, and believe the following solution might be the most appropriate.

For tumor growth curve:

Student's t-test for two groups;

ANOVA for three or more groups (one-way or two-way, depending on the variables).

For bar graphs:

Student's t-test for two groups;

ANOVA for three or more groups (one-way or two-way, depending on the variables).

A simple professional explanation for our selection can be found from the link below:

<https://www.scribbr.com/statistics/statistical-tests/>

We have corrected all the related graphs, and all test conclusions remain unchanged.

11. Finally, the authors used numerous misleading or confusing words, phrases or terms throughout the manuscript. For example, the phrase “blockade therapy” used in many places including subheadings, which however has no bearing at all in this study that is restricted to mice. What does “immunogenic apoptosis” mean? “...products of OXPHOS, such as ATP and ROS, have been shown to promote anti-tumor responses”: ATP can promote tumor growth, via activation of the P2X7 receptor in tumor cells and at the same time inhibit tumor growth by activation of the P2X7 receptor in tumor-residing immune cells! In addition, the authors failed to provide the reference or evidence to support the statements in many parts, e.g., “Extracellular ATP release is actively regulated via autophagy upon chemical drug treatment”, “IFN-Is induce autophagy and ATP release in tumor cells”, “maintaining a high ATP concentration by preventing its degradation into immunosuppressive adenosine may facilitate anti-tumor immunity induced by CD47 blockade”, “We also found that CD47-SIRP α ICB efficacy depended on IL-1beta”?

Response:

We are sorry for these misleading words. We have carefully examined and corrected.

More specifically:

- “blockade therapy”: we clarified this is in mouse study at first appearance and whenever possible.
- “immunogenic apoptosis”: we cited some important papers for further informaiton.
- “tumor-promoting role of ATP”: The complicated role of ATP, together with other

OXPHOS products, have been systemically discussed, which is also raised by Reviewer 4, point 3.

- Literature citation has been carefully added.

Reviewer #3 (Remarks to the Author): with expertise in cancer immuno-metabolism

Comments to:

Tumor cell intrinsic type I interferon signaling mediated metabolic remodeling dictates CD47-SIRPa blockade immunotherapy

Zhou et al. for Nature Communications

Summary

In this manuscript, Zhou et al elaborates on the mechanistic understanding of CD47-SIRPa immune blockade therapy. Using RNA-seq on tumor cells exposed to SIRPa-hIg (CV-1) they find that IFN-I signaling, and autophagy is upregulated after CV-1. They delineate a pathway where IFN-I activates IFNAR and Isg15 on tumor cells, resulting in increased OXPHOS and autophagy further leading to release of ATP. Using CRISPR/Cas9 guided KO of tumor cells, they show that tumor cells without Isg15, Ndufs6 (OXPHOS) or Atg5 (Autophagy) are unresponsive to CV-1 therapy.

Zhou et al argue that CV-1 promotes T cell priming through this mechanism, since injecting CV-1 in IFNAR^{-/-} tumors did not have an effect on IFNAR^{-/-} or a distant WT tumor in the same mouse, whereas injecting CV-1 into WT tumor, suppressed distant IFNAR^{-/-} tumors in the same mouse, and CV-1 increased CD8⁺ T cell percentage in an IFNAR-dependent way. They use mixed bone marrow chimera of ATP receptor P2X7R^{-/-} and CD11c-DTR mice to show that mice without P2X7R in DCs become resistant to CV-1 treatment, and blocking IL-1b with IL-1Ra abolished CV-1 effect.

Lastly, they show that overexpression of ATP degrading enzyme CD39 by tumor cells, abolishes the effect of CV-1 on tumor growth, whereas CD39 inhibitor or CD73 antibody enhanced anti-tumor efficacy in combination with CV-1, supporting that high extracellular ATP facilitate anti-tumor response by CD47 blockade.

This work presents novel tumor cell intrinsic mechanistic findings that expands our understanding of CD47-SIRPa blockade therapy using thorough experimental approaches. A few limitations can be found in the immunological investigations which are listed below for further revision.

Major comments

- 1. In this manuscript they show that IFN-I induces OXPHOS and autophagy, which are both required for ATP release. Yet, how these two are associated is not clear. Could the authors show if mitophagy is also regulated in this setting, and if that is relevant for IFN-I-induced ATP release?**

Response:

Thank you for the overall positive comment. As to this specific point, it is interesting

to link them together. We have used a classical method (mKeima Assay for Mitophagy) to test whether IFN-I would increase mitophagy. The Keima fluorophore has bimodal excitation under neutral and acidic pH conditions. When Keima is targeted to mitochondria it can accurately reveal the formation of mitolysosomes. (Curr Protoc Cell Biol. 2020 Mar;86(1):e99.) The data showed that IFN-I does not influence mitophagy at all, while the positive control CCCP does significantly (Figure below). These data suggest at least two points: first, although mitochondrial OXPHOS is increased, mitochondria is still largely healthy, therefore no mitophagy; second, we do not need to worry mitophagy to inhibit OXPHOS and consequent ATP release and anti-tumor effect, at least in this condition. However, whether mitophagy would be induced in vivo during prolonged CV-1 treatment is still an open question, remaining to be investigated in future.

- The authors claim that tumor cell intrinsic IFN-I signaling is required for T cell priming after CV-1, yet this is only supported by increased CD8+ T cell percentages in the tumor. To confirm that this is due to priming other experiments are needed. For example by ex vivo co-culture assays with CV1/ATP-treated APCs.

Response:

Thank you for this important advice. We have performed this experiment as the reviewer suggested. C57BL/6 mice were inoculated with MC38-OVA cells. Mice were injected with 50 µg hIgG or CV-1 every three days. Two days after the third injection, DCs were sorted from tumors and incubated with OT-I T cells in 1:10 ratio. OT-I cell activation were detected 24h after incubation by FACS analysis of CD69 and CD62L expression. The data showed that DCs isolated from CV1-treated MC38-OVA tumors showed a stronger ability to activate naïve OT-I T cells than those isolated from control hIgG treated tumors. (Fig. 8d)

3. CD8 T cell percentages are increased, but the functionality of the T cells is not investigated. Killing assay or cytokine release assays of CD8 T cells is necessary to state that they are increased in functionality. Note: In the result section it is written “IFNAR1 deficiency in tumor cells does not influence the effector T cell killing function”, yet without any figure reference or reference to literature.

Response:

We thank for this important point. We have performed this experiment as the reviewer suggested. C57BL/6 mice inoculated with WT MC38 cells were treated i.t. with 50 μ g CV-1 or hlgG every three days. Two days after the third treatment, CD45+ cells were sorted from tumors and stimulated with MC38 tumor cell lysis, tumor-specific IFN- γ cells were measured by the ELISpot assay. The results showed that CD45+ cells isolated from CV1-treated tumors enriched more more IFN- γ -producing cells (Fig. 2c). Moreover, to further detect IFN- γ -producing T cells, we adopted the MC38-OVA tumor model and OT-I peptide was used for in vitro restimulation. The results showed that CV-1 treatment induced more IFN- γ -producing CD8+ T cells than control treatment (Fig. 2d).

The figure panel and a reference literature were cited for the sentence “IFNAR1 deficiency in tumor cells does not influence the effector T cell killing function”. The description of this result was also rephrased for better understanding.

Minor comments

1. **IFN-I increases both mitotracker green (MDR) and mitotracker deep red (MDR) suggesting an increased mitochondrial biosynthesis. To give an expression of the qualitative differences of mitochondria, the authors should also show mitochondrial functionality normalized to mitochondria mass (MDR/MG).**

Response:

The ratio of MDR to MG, an indication of mitochondrial activity per mitochondrial mass, was decreased with IFN- α stimulation (shown below), suggesting that tumor cells might fail to fully utilize mitochondrial activity in IFN- α stimulation, which requires further attention for improved efficacy. These data have been added as Fig. 3c and Supplementary Fig. 3c.

2. **In figure 4e it looks like ISG15 KO MC38 grows slower than WT, possibly explaining why the responsiveness is ablated in these tumors. Can the authors comment in this?**

Response:

The genetic deletion of *Isg15* in tumor cells slows down the tumor growth in vivo. Whether tumor growth rate may correlate with therapeutic responsiveness is not certain. For example, in our IFNGR-KO MC38 tumors, which grow much slower than WT MC38, is highly responsive to CD47 blockade therapy (Fig. 1e). Thus, this might be a case by case issue. Carefully examination is required for further dissecting the relationship. Since we always compare CV1 treatment with control hIgG treatment for both WT and KO tumors side by side, there should be no problem to draw our conclusion that *Isg15* deficient tumors are resistant to CV1 treatment.

3. **CD8 T cell percentages are increased in tumors after CV-1 injection, however the count of CD8 T cells normalized to tumor weight is also relevant to show.**

Response:

We are sorry the tumor weight was not always measured, which would be more accurate to calculate the density of CD8+ T cells. However, since the CD8+ T cell percentage shown in Fig. 8b-c is a percentage of total cells in tumor tissues (including

both tumor cells and non-tumor cells), this would be approximate to the density. We are sorry the y-axis label is not clear, and have corrected.

Reviewer #4 (Remarks to the Author): with expertise in cancer, CD47

Tumor metabolic rewiring plays a critical role in cancer immunotherapy, however, some details in the metabolic features in the tumor microenvironments remain uneducated. In this study, Zhou et al investigated IFN-Is, pleiotropic cytokines enriched in the tumor microenvironment, in tumor metabolic rewiring with OXPHOS-mediated resistance to anti-CD47 therapy. The results indicate that tumor responsiveness to IFN-Is is essential for CD47-SIRP α blockade immunotherapy; IFN-Is can guide the tumor metabolic rewiring with OXPHOS activation that enhances cellular ATP generation which together with autophagy-derived ATP can raise the extracellular ATP pool leading to enhanced tumor cell response to anti-CD47 therapy. ATP released upon CD47-SIRP α blockade primes the anti-tumor T cell response via ATP-P2X7 receptor-mediated dendritic cell activation which is supported by ATP-degrading ectoenzymes, CD39 and CD73. These results reveal a new mechanism of enhanced OXPHOS that can boost tumor response to CD47-targeted ICT.

The overall experimental design is reasonable, and the major data are supportive of the conclusion which could contribute to a new version of metabolism-associated tumor response to ICT, thus holding a potential clinic benefit. However, in addition to scientific deficiency including the lack of macrophagic analysis, there are some major concerns about professional data presentation and explanation. The overall scientific writing is to be substantially improved. A focus and deepening the analysis of the diet using the in vivo test would be appreciated and informative.

Scientific Comments:

- 1. The specific pathways of IFN-Is activate tumor OXPHOS which are believed to be deficient in many solid tumor cells (Warburg Effect). Does shifting from glycolysis to OXPHOS dominant pathway occur in the tumor model used in this work?**

Response:

Thank this reviewer for the overall positive comments. As to the metabolism shift, it is an interesting topic. We have examined the tumor cell metabolism in vitro using Seahorse assay. We found that upon IFN-I treatment, MC38 tumor cells showed increased OXPHOS (Fig. 3c), while glycolysis was not inhibited (Supplementary Fig. 3h). As we described in the Introduction part, although tumor cells usually demonstrated as a predominant Warburg effect, and reduced mitochondrial OXPHOS, this is not due to permanent impairment of mitochondrial OXPHOS, but partly due to suppression by enhanced glycolysis. This means tumor cell metabolism is plastic and can be further reprogrammed in response to external stimuli. Here, we found that IFN-I is just such a signal that can upregulate tumor cell OXPHOS. Although we did not find glycolysis inhibition in our in vitro assay, it would be interesting to test in future whether prolonged IFN-I signal would tune down glycolysis at later time point.

This has been briefly discussed in association with the corresponding results.

- 2. It is not clear whether tumor resistance to CD47-SIRPa blockade is due to the reduced or enhanced copy numbers of CD47 receptors. Does the CD47 gene copy number enhanced or reduced by mitochondrial OXPHOS?**

Response:

This is an interesting question. Altered CD47 copy number in ovarian serous cystadenocarcinoma was reported (PMID: 27569584). Since protein expression on tumor cell surface is the functional form of CD47, it might be more meaningful to examine CD47 protein expression. Therefore, we measured CD47 expression on WT, IFNAR1-KO and IFNGR-KO MC38 tumor cells. We found that IFNAR1-KO tumor cells have slightly increased CD47 expression than WT cells, and IFNGR-KO tumor cells have more increase (shown below). This trend of CD47 expression level does not correlate with their responsiveness upon CV-1 treatment, since both WT and IFNGR-KO tumors are responsive to CV-1 treatment while IFNAR1-KO tumors are not. Thus, we do not think the tumor resistance to CD47-SIRPa blockade is solely due to altered CD47 expression. To what extent the CD47 expression level influences the blockade efficacy needs more careful designs.

As to the effect of OXPHOS on CD47 expression, it is difficult to specifically manipulate OXPHOS in our current model, but we have found upon CD47-SIRPa blockade treatment in vivo, there is no altered CD47 expression change on WT MC38 tumor cells (shown below).

- 3. In addition to ATP generation and release, OXPHOS can produce many other biological effects which can alter cell survival rate with or without ICT, which should be discussed.**

Response:

In addition to ATP, ROS is an important byproduct of OXPHOS. ROS has complicated effects on tumor cells. At low to moderate concentrations, ROS directly promotes activation of multiple pathways leading to tumor cell proliferation, including CDK2 [5], HIF1a [6], PI3K/AKT/mTOR [7-9], and MAPK/ERK [10-12]. On the opposite, exceeding ROS promotes tumor cell programmed cell death, via apoptosis [13, 14], necroptosis [15], and ferroptosis [16]. In addition, ROS may influence tumor growth indirectly via immune cells [17, 18]. The direct effect of ROS on tumor cell proliferation or death unlikely contributes to tumor growth inhibition in the CD47-SIRP α blockade scenario, since without T cell-mediated immune response, the blockade cannot inhibit tumor growth at all (Fig. 2a,b). However, we cannot exclude the possibility that ROS-mediated tumor cell death may activate immune cells for an enhanced antitumor response. The exact role of tumor ROS in CD47-SIRP α blockade remains to be studied in future.

This discussion has been added into the text.

- 4. In Fig. 1, it is unclear why need to sort the tumor cells after in vivo tumor treatment with CD47-SIRP α blockade, and what is the control transcriptomics compared in the treated tumor cells.**

Response:

To specifically examine the response of tumor cells themselves to this treatment, we therefore sorted the tumor cells to reduce host cell contamination for transcriptome analysis. Tumor cells from hIgG treated mice were used as a control.

Scientific Presentation

- 5. The overall writing, especially on the precise description of experimental procedures is in need; especially a critical review by senior authors is mandatory (Dr. YX Fu, a well-known expert in the field is listed as one of the authors but not involved in the paper writing).**

Response:

We are sorry for the writing issues. The whole text has been carefully examined and further reviewed by Dr. Fu.

- 6. In the introduction, critical reading is required for improving the scientific language and paper citation, such as “In recent years, the innate immune checkpoint has emerged as a popular target for cancer immunotherapy in recent years. The CD47-SIRP α axis is the best-studied innate checkpoint in cancer. Therapies designed to disrupt the CD47-SIRP α interaction are by far the furthest along the clinical development process and are now being investigated for multiple human cancers”. This description does not reflect the current anti-CD47 and is not precisely related to the PD-1/PD-L1 blockage therapy.**

Response:

We are sorry for the inadequate description. We have extensively update this paragraph. Just recently, a late clinical trial of a CD47-blocking antibody was discontinued, and some other trials were also restrained. This situation was also updated in this paragraph.

In addition, we have no intent to link CD47-SIRP α blockade therapy to PD-1/PD-L1 here. The description in this paragraph focuses on innate checkpoint.

- 7. As cited in ref 48, Isg15 has been reported to govern mitochondrial function in macrophages following vaccinia virus infection (ref 48) which found that “Post-translational modifications such as ISGylation regulate essential mitochondrial processes including respiration and mitophagy, and influence macrophage innate immunity signaling”. A further digger into the Isg15-mediated specific elements in the mitochondria respiration such as the complexes phosphorylation would be informative.**

Response:

The role of ISG15 in OXPHOS has been reported by several studies. In addition to the one we already cited [19], some other groups have also reported similar findings in different cell types, even including tumor cells.

In the original paper [19], Baldanta et al. reported, in BMDM treated with IFN-I, that both basal and maximal oxygen consumption rate (OCR) were reduced in the absence of ISG15 as measured by Seahorse. This is associated with reduced ISGylation of mitochondrial proteins.

Sonia et al. reported similar finding [20], in pancreatic cancer stem cells, that ISG15 loss results in decreased ISGylation of mitochondrial proteins concomitant with increased accumulation of dysfunctional mitochondria and reduced spare respiratory capacity (SRC).

In another study [21], Waqas et al. reported that ISG15-deficient microphages have

also reduced mitochondrial respiration as measured by Seahorse, which is associated with reduced glutamine uptake and reduced expression of nuclear genes encoding components of mitochondrial respiratory chain complexes II-V.

According to these studies, it seems that ISG15 might influence OXPHOS in both direct manners, via ISGylation of mitochondrial proteins for instance, and indirect manners, via controlling glutamine uptake or OXPHOS-related gene expression. These mechanisms are not necessarily mutually exclusive. The exact mechanism in our model still remains to be clarified. We think it might be out of the scope of current study. We hope we can leave this question open. But we have cited these studies with brief discussion.

- 8. The first sentence of the Results “To investigate tumor cell responsiveness during immunotherapy (what is the specifically immunotherapy indicated here?), established MC38 tumors (what are the tumors used here, should be indicated here) were treated with SIRP α -hIg (CV-1) (what is this, antibody?), a high-affinity mutant for CD47 binding”.**

Response:

We are sorry for these inadequate descriptions. We have corrected them for clearer presentation.

References:

1. Li, X.-Y., et al., *Targeting CD39 in Cancer Reveals an Extracellular ATP- and Inflammasome-Driven Tumor Immunity*. *Cancer Discovery*, 2019. **9**(12): p. 1754-1773.
2. Qin, J., et al., *Blocking P2X7-Mediated Macrophage Polarization Overcomes Treatment Resistance in Lung Cancer*. *Cancer Immunology Research*, 2020. **8**(11): p. 1426-1439.
3. Casey, M., et al., *Inhibition of CD39 unleashes macrophage antibody-dependent cellular phagocytosis against B-cell lymphoma*. *Leukemia*, 2023. **37**(2): p. 379-387.
4. Adinolfi, E., et al., *Accelerated Tumor Progression in Mice Lacking the ATP Receptor P2X7*. *Cancer Research*, 2015. **75**(4): p. 635-644.
5. Kirova, D.G., et al., *A ROS-dependent mechanism promotes CDK2 phosphorylation to drive progression through S phase*. *Dev Cell*, 2022. **57**(14): p. 1712-1727.e9.
6. Movafagh, S., S. Crook, and K. Vo, *Regulation of hypoxia-inducible factor-1 α by reactive oxygen species: new developments in an old debate*. *J Cell Biochem*, 2015. **116**(5): p. 696-703.
7. Satooka, H. and M. Hara-Chikuma, *Aquaporin-3 Controls Breast Cancer Cell Migration by Regulating Hydrogen Peroxide Transport and Its Downstream Cell Signaling*. *Mol Cell Biol*, 2016. **36**(7): p. 1206-18.
8. Salmeen, A., et al., *Redox regulation of protein tyrosine phosphatase 1B involves a sulphenyl-amide intermediate*. *Nature*, 2003. **423**(6941): p. 769-73.
9. Lee, S.R., et al., *Reversible inactivation of the tumor suppressor PTEN by H₂O₂*. *J Biol Chem*, 2002. **277**(23): p. 20336-42.
10. Liu, H., et al., *Activation of apoptosis signal-regulating kinase 1 (ASK1) by tumor necrosis factor receptor-associated factor 2 requires prior dissociation of the ASK1 inhibitor thioredoxin*. *Mol Cell Biol*, 2000. **20**(6): p. 2198-208.
11. Burgoyne, J.R., et al., *Cysteine redox sensor in PKG1 α enables oxidant-induced activation*. *Science*, 2007. **317**(5843): p. 1393-7.
12. Kamata, H., et al., *Reactive oxygen species promote TNF α -induced death and sustained JNK activation by inhibiting MAP kinase phosphatases*. *Cell*, 2005. **120**(5): p. 649-61.
13. Kagan, V.E., et al., *Cytochrome c acts as a cardiolipin oxygenase required for release of proapoptotic factors*. *Nat Chem Biol*, 2005. **1**(4): p. 223-32.
14. Zuo, Y., et al., *Oxidative modification of caspase-9 facilitates its activation via disulfide-mediated interaction with Apaf-1*. *Cell Res*, 2009. **19**(4): p. 449-57.
15. Zhang, Y., et al., *RIP1 autophosphorylation is promoted by mitochondrial ROS and is essential for RIP3 recruitment into necrosome*. *Nat Commun*, 2017. **8**: p. 14329.
16. Xie, Y., et al., *Ferroptosis: process and function*. *Cell Death Differ*, 2016. **23**(3): p. 369-79.

17. Liu, R., et al., *Oxidative Stress in Cancer Immunotherapy: Molecular Mechanisms and Potential Applications*. *Antioxidants*, 2022. **11**(5): p. 853.
18. Kotsafti, A., et al., *Reactive Oxygen Species and Antitumor Immunity—From Surveillance to Evasion*. *Cancers*, 2020. **12**(7): p. 1748.
19. Baldanta, S., et al., *ISG15 governs mitochondrial function in macrophages following vaccinia virus infection*. *PLOS Pathogens*, 2017. **13**(10): p. e1006651.
20. Alcalá, S., et al., *ISG15 and ISGylation is required for pancreatic cancer stem cell mitophagy and metabolic plasticity*. *Nature Communications*, 2020. **11**(1): p. 2682.
21. Waqas, S.F.-u.-H., et al., *ISG15 deficiency features a complex cellular phenotype that responds to treatment with itaconate and derivatives*. *Clinical and Translational Medicine*, 2022. **12**(7): p. e931.

REVIEWER COMMENTS

Reviewer #1 (Remarks to the Author):

The authors adequately addressed the points I raised, except for the functional data related to the knock out of *Ifngr1* and *Ifnar1*. The flow cytometry data for pSTAT1 are not very convincing. Western blot data should give a clearer result, if IRF1 (for *Ifngr1* ko) and (p)IRF3 (for *Ifnar1* ko) would be included. Such Western blot data should also be presented for the additional tumor models.

Reviewer #2 (Remarks to the Author):

The authors have made considerable efforts in revising the manuscript, including new data. Nonetheless, as readily noticed in their point-to-point responses, a number of questions of high importance or relevance to this study raised by the reviewers are not dealt with adequately or not at all.

In addition, the explanation for the reasons for the high variations in data from experiments performed at different times or by different researchers is understandable, but not acceptable as the readers cannot discriminate or have the opportunity to know the exact origins or nature of such variations and, more importantly, whether such variations reflect the true experimental errors, related to reproducibility. It is the responsibility of the authors to provide high quality data to avoid misleading or misinterpretation.

Finally, as clarified in authors' response: "n = 3 refers to three biological replicates from one representative experiment. Each experiment was repeated at least twice. Data are representative of at least two independent experiments." This practice is totally abusing statistical analysis, or inadequate, saying the least, if authors have some basic knowledge of statistics! The data from biological replicates are prone to missing the important experimental errors (randomly occurring and related to data reproducibility) and carrying uncorrectable experimental errors (e.g., arising from using poorly calibrated tools or equipment), which represents a very common risk or reason for drawing false conclusions. Therefore, n = 3 is meant to indicate three independent experiments, which provides the

minimal requirements to infer the data (error) distribution (albeit barely possible for $n = 3$, the most common or typical abuse of statistics by biologists) and to choose the statistical analysis method accordingly. The authors need to present each data points from individual independent experiments rather than biological replicates, where applicable.

Reviewer #3 (Remarks to the Author):

The authors address most of my concerns. However, the following two points should be better addressed.

1. The authors didn't answer our request for T cell counts, since they don't have the tumor weights. If they want to claim that T cell infiltration is higher, this is needed.

To support their claim that CV-1 treatment enhances DC priming of T cells, they performed ex vivo co-culture of tumor-associated DCs with naïve OT1s. However, the results are super weak. They find an increase CD69+ OT1s (from 5% to 7.5%) and CD62L- OT1s (From 1.8% to 2.2%) after 24h. It does not seem convincing (Fig 8d).

2. It might not be priming that is enhanced but just antigen presentation within the tumor sustaining the T cells, but they still haven't showed anything else than DC KO of ATP-receptor reduce response to therapy.

Reviewer #4 (Remarks to the Author):

The authors have thoroughly addressed the enquires regarding the data presentation and the scientific presentation has been improved. The finding of mitochondrial OXPHOS enhancement in tumor resistant to CD47 checkpoint therapy holds the novelty and clinical relevance.

Dear Reviewers,

We appreciate that most of our revisions have been accepted by the reviewers. For a few remaining concerns, we have addressed as below. We hope the current manuscript is suitable for publication. Thank you.

Reviewer #1 (Remarks to the Author): with expertise in cancer immunology, IFNs

The authors adequately addressed the points I raised, except for the functional data related to the knock out of *Ifngr1* and *Ifnar1*. The flow cytometry data for pSTAT1 are not very convincing. Western blot data should give a clearer result, if IRF1 (for *Ifngr1* ko) and (p)IRF3 (for *Ifnar1* ko) would be included. Such Western blot data should also be presented for the additional tumor models.

Response:

We appreciate this reviewer's recognition on our revisions. As to the functional confirmation of cell line knockout efficiency, we agree it is important. Following this reviewer's suggestion, we determined pSTAT1 by Western blot; indeed, it is more convincing (Supplementary Fig. 3c and also shown below). This strongly confirmed that for all three types of cell lines, MC38, CT26 and A20, both IFNRA1 KO and IFNGR1 KO are functionally deficient in response to IFN- α or IFN- γ stimulation, respectively.

The reviewer also suggested to test IRF1 and (p)IRF3 for further confirmation. IRF1 is a typical IFN-g inducible gene [1]. However, (p)IRF3 is not typical downstream of IFNAR1 signaling. It is usually activated by various pattern recognition receptors but not type I interferons [2, 3]; in fact, (p)IRF3 is upstream of type I interferons since it induces their expression [3]. Thus, we agree IRF1 is a good readout for confirmation of *Ifngr1* ko efficiency, while (p)IRF3 may not be for *Ifnar1* ko. Therefore, we determined IRF1 induction by Western blot (Supplementary Fig. 3c), and also by RT-PCR (Supplementary Fig. 3d and shown below). In addition, we also tested some typical ISGs by RT-PCR, including Mx1 (downstream of both IFNRA1 and IFNGR1) and Ido1 (specifically downstream of IFNGR1).

d
Reviewer #2 (Remarks to the Author): with expertise in cancer, ATP

- **The authors have made considerable efforts in revising the manuscript, including new data. Nonetheless, as readily noticed in their point-to-point responses, a number of questions of high importance or relevance to this study raised by the reviewers are not dealt with adequately or not at all.**

Response:

Thank for this reviewer for the recognition of our efforts. We have tried our best and properly addressed the major concerns of all reviewers. Although we mentioned a few “remain to do in future” issues, we believe those are not major concerns and would not influence the significance, the novelty and the major mechanisms. No study is perfect, something “remained to do” always exist. So is the current one and also many other published papers. This is why many papers published nowadays include a specific section of “limitations of the study”. We think this publication policy is to foster, in the scientific community, the communication of values even it has some minor limitations. We hope this is acceptable.

- **In addition, the explanation for the reasons for the high variations in data from experiments performed at different times or by different researchers is understandable, but not acceptable as the readers cannot discriminate or have the opportunity to know the exact origins or nature of such variations and, more importantly, whether such variations reflect the true experimental errors, related to reproducibility. It is the responsibility of the authors to provide high quality data to avoid misleading or misinterpretation.**

Response:

We appreciate the understanding of this reviewer. However, we would like to mentioned that the variation between experiments are not so big as the reviewer previously pointed out, particularly after correction or normalization which we missed in the first version. An exception is extracellular ATP detection in Fig. 4d and Supplementary Fig. 5c, in which the eTAP levels are about 10 folds lower than those in other experiments. Since Isg15 deficient cells were more sensitive to IFN- α treatment, cells were stimulated for 24 hours, instead of 48 hours, before detection to avoid cell death in these assays. We are sorry for not making this clear before and have clarified this in the revision.

- **Finally, as clarified in authors’ response: “n = 3 refers to three biological replicates from one representative experiment. Each experiment was repeated at least twice. Data are representative of at least two independent experiments.” This practice is totally abusing statistical analysis, or inadequate, saying the least, if authors have some basic knowledge of statistics! The data from biological replicates are prone to missing the important experimental errors**

(randomly occurring and related to data reproducibility) and carrying uncorrectable experimental errors (e.g., arising from using poorly calibrated tools or equipment), which represents a very common risk or reason for drawing false conclusions. Therefore, $n = 3$ is meant to indicate three independent experiments, which provides the minimal requirements to infer the data (error) distribution (albeit barely possible for $n = 3$, the most common or typical abuse of statistics by biologists) and to choose the statistical analysis method accordingly. The authors need to present each data points from individual independent experiments rather than biological replicates, where applicable.

Response:

We are sorry we cannot agree on this point. Although the method described by the reviewer is definitely correct, it does not mean other methods are wrong. In fact, method such as “ $n = 3$ refers to three biological replicates from one representative experiment” is broadly used in most papers, which can be easily seen from randomly picked papers published on even high-profile journals. This is also in line with the reproducibility policies of Nature Communications, as can be seen from the link (<https://www.nature.com/articles/s41467-018-06012-8>). In addition, we have used “biologically independent samples” rather than “replicates” to avoid misunderstanding according to the policy.

Reviewer #3 (Remarks to the Author): with expertise in cancer immuno-metabolism

The authors address most of my concerns. However, the following two points should be better addressed.

- **The authors didn't answer our request for T cell counts, since they don't have the tumor weights. If they want to claim that T cell infiltration is higher, this is needed. To support their claim that CV-1 treatment enhances DC priming of T cells, they performed ex vivo co-culture of tumor-associated DCs with naïve OT1s. However, the results are super weak. They find an increase CD69+ OT1s (from 5% to 7.5%) and CD62L- OT1s (From 1.8% to 2.2%) after 24h. It does not seem convincing (Fig 8d).**

Response:

Thank this reviewer for the recognition of our revisions.

As to the T cell count issue, we agree it would not be accurate to claim higher infiltration, although the percentage may indicate so (Please pay attention that the percentage here is CD8+ T cells/total cells of the tumor tissue, including tumor cells, hematopoietic cells and other stromal cells). To avoid misleading and to be more accurate, we described this situation in the text and also changed the word “infiltration” to “percentage”.

As to the DC priming efficacy, we agree this difference is not huge, but it is repeatable with statistical significance. In addition, specifically for CD69 upregulation, a typical marker for naïve T cell activation, a change from 5% to 7.5% is a 50% increase relatively speaking. We think the reason for the low percentage of T cell activation is probably due to the limitation of the assay. MC38-OVA tumor model was used here, in which tumor cell-endogenous OVA antigen expression, release, capture, processing and presentation by tumor-associated DCs might not be so prevalent that can be easily detected. We mentioned this limitation in the text and hope this limitation would not obscure the overall value of this study, which way is commonly appreciated in many journals nowadays as shown by “limitations of the study”. We hope this is acceptable.

- **It might not be priming that is enhanced but just antigen presentation within the tumor sustaining the T cells, but they still haven't showed anything else than DC KO of ATP-receptor reduce response to therapy.**

Response:

This is probably due to different understanding about the word “priming” between the reviewer and us. The reviewer uses the word “priming” in a stricter way meaning naïve T cell activation by APCs, while we used it more loosely referring to T cell response

induction. We agree strict use of the word would avoid misunderstanding. Therefore, we changed the phrase “T cell priming” to “T cell response induction”, except for the in vitro DC-T activation assay, in which true T cell priming is evaluated. Thus, three lines of evidences will support the role of CV-1 treatment in T cell response induction: 1) bilateral tumor inoculation model; 2) upregulated expression of costimulatory molecule CD80 on DCs, 3) enhanced in vitro priming by DCs, and 4) impaired efficacy upon cKO of P7X7R on DCs.

References:

1. Honda, K. and T. Taniguchi, *IRFs: master regulators of signalling by Toll-like receptors and cytosolic pattern-recognition receptors*. Nature Reviews Immunology, 2006. **6**(9): p. 644-658.
2. Yanai, H., et al., *Revisiting the role of IRF3 in inflammation and immunity by conditional and specifically targeted gene ablation in mice*. Proc Natl Acad Sci U S A, 2018. **115**(20): p. 5253-5258.
3. Al Hamrashdi, M. and G. Brady, *Regulation of IRF3 activation in human antiviral signaling pathways*. Biochemical Pharmacology, 2022. **200**: p. 115026.

REVIEWERS' COMMENTS

Reviewer #1 (Remarks to the Author):

The authors addressed the points that I raised.